# Expectation of the maximum of random variables with applications to reinforcement learning

## Abstract

We explore the application of expressions for the expected maxima of sets of random variables to compute the mean of the distribution of fixed points of the Bellman optimality equation for large state space Markov decision processes (MDPs), under a Bayesian framework. Current approaches to computing the statistics of the value functions in reinforcement learning rely on bounds and estimates that do not exploit statistical properties of the Bellman equation which can arise in the large system limit. Specifically, we utilise a recently developed mean field theory called *dynamic mean field programming*, to compute the moments of the value function. Under certain conditions on the MDP structure and beliefs this mean field theory is exact. Computing the solution to the mean field equations however relies on computing expected maxima, and previous approaches were limited to identically distributed rewards. We improve the existing estimates and generalise to non-identical settings. We analyse the resulting approximations to the mean field equations, establishing Lyapunov stability and contractive properties.

## 1 Introduction

An outstanding challenge in Bayesian reinforcement learning is to accurately compute or approximate the posterior distribution of solutions to the Bellman optimality equation, known as the optimal value functions. In principle, with the posterior one can then design sample efficient learning algorithms, in the style of so-called multi-armed bandit problems (Lattimore and Szepesvári, 2020).

Index policies provide excellent solutions for the multi-armed bandit, a Markov decision process (MDP) with a single state and a number of actions, each of which returns a random reward. An index represents the benefit of choosing a particular action at a particular time, given observed rewards. A good index policy balances, in expectation, the reward one might receive with the information one gains by choosing a particular action. Sample efficient algorithms act by choosing the action with the highest index, for example, the highest upper confidence on the mean-reward for each action, a principled heuristic known as the "optimism in the face of uncertainty".

A prominent example in the frequentist setting are the upper confidence bound algorithms (Lattimore and Szepesvári, 2020), which have an analogue in the Bayesian approach (Kaufmann et al., 2012). Often the index has the form of a mean estimate, plus a constant multiplied by the standard deviation in the estimate, where the constant scales inversely with the number of times the action has been selected.

Extending this approach to general MDPs is difficult because the quantity that corresponds to a mean-reward in the bandit setting is the *optimal state-action value function*, also known as *optimal Q-values*. The problem is that one can not obtain direct samples to estimate these functions (even if one had the optimal policy!). As (O'Donoghue et al., 2018) describe,

> "If we had access to the true posterior distribution over the Q-values then we could take actions that lead to states with higher uncertainty by, for example, using Thompson sampling (Thompson, 1933); (Strens, 2000), or constructing Q-values that are high probability

> upper bounds on the true Q-values and using the [optimism in the face of uncertainty] principle (Kaufmann et al., 2012). However, calculating the true posterior is intractable for all but very small problems. Due to this difficulty prior work has sought to approximate the posterior distribution (Osband and Van Roy, 2017), and use that to drive exploration."

Our paper takes this statement as a challenge and our starting point. We attempt to estimate the moments of the *optimal* Q-values directly, beginning with the first moment, which itself presents difficulty and also considerable insight into how uncertainty propagates. In contrast to existing methods, we estimate the mean using a recently developed mean field theory for reinforcement learning (Stamatescu, 2022).

## 1.1 New approaches to quantifying uncertainty over the optimal Q-values

Although one can not sample the optimal state-action values directly, as an agent interacts with an MDP, one does observe directly the rewards and state transitions and can then estimate the MDP parameters. Thus, current approaches to deriving an index over optimal Q-values are based on translating confidence sets or Bayesian credibility sets for the MDP parameters, to confidence intervals on the state-action values.

This translation is generally hard due to the interdependence induced by the Bellman optimality equation, as described in the quote from (O'Donoghue et al., 2018) above. Moreover, methods for translating these sets to the state-action value functions are generally evaluated via a regret bound, that is, at the end of an asymptotic analysis of the learning algorithm. In practice however, such translations have result in both biased estimates for the mean and in loose confidence intervals for the state-action value functions (Osband and Van Roy, 2017).

A recent approach to this problem introduces a statistical mean field theory for Bayesian reinforcement learning called dynamic mean field theory (DMFT), (Stamatescu, 2022). The DMFT results in a set of *mean field equations* dubbed *dynamic mean field programming* (DMFP), that propagate parameter uncertainty through the Bellman optimality equation exactly, under certain conditions.

The DMFP approach is based on a entirely different method of analysis, which was shown to calculate *exactly* the mean and higher moments of the state-action value functions, under certain assumptions. If we weaken the assumptions, the DMFP equations become an approximation, which may still be applied with success.

The mean field or DMFP equations, as the name suggests, propagate the *moments* through the Bellman optimality equation, or dynamic programming equations. In the finite horizon case, these are the moments of the time-dependent value functions, and in the infinite horizon case, these are the moments of the *iterates* of the Bellman equation, whose fixed point is the optimal value function.

This idea is analogous, in a certain way, to message passing algorithms for approximate inference, such as belief propagation[1]. This connection was not made explicitly in the work of (Stamatescu, 2022), which focused on the identical prior case where there is one representative mean field equation for all state-action values. The present paper improves the computation of the existing mean field equations and extends them to the case of non-identical beliefs.

Explicitly, in order to solve the mean field equations, one must compute expectations over the maximum of the state-action value functions. The approach presented in (Stamatescu, 2022) applies an extreme value theory approximation, and is thus restricted to the large action space limit and the identical reward case. The task of computing the expected maxima under general conditions is the starting point for the present paper, from which our main contributions follow.

## 1.2 Summary of the contributions

The current paper forms crucial building blocks for new methods to quantify uncertainty over state-action value functions in Bayesian reinforcement learning (RL). Here we tackle head on the problem of estimating the moments of the state-action value functions, by improving the computation of mean field approximations.

---

[1]Precisely, the dynamic mean field theory is analogous to the *average case* analysis of *dynamic* message passing algorithms for an inference problem.

We focus on the mean of the optimal Q-values, which itself is difficult to estimate. While the prior work of (Stamatescu, 2022) focused on the case of identical beliefs over the MDP parameters, we study the mean field equation for general non-identical beliefs.

In the statistical inference language, the mean field equations are equivalent to so-called *message passing* algorithms, albeit for dynamic random variables. Message passing algorithms such as belief propagation or expectation propagation allow one to compute the marginal moments of unobserved or latent variables (which are static, having no dynamic time index). Such an approach has not been studied in Bayesian reinforcement learning, to the best of our knowledge.

A summary of our contributions are as follows,

- (Review and numerical study of the expected maximum)

  We provide a comprehensive review of disparate exact formulae, approximations and bounds of the expected maximum, largely from the statistics literature. We then numerically investigate the reviewed formulae, thus providing the basis for the efficient and accurate computation of new mean field equations for uncertainty propagation in reinforcement learning.

- (Bayesian RL and uncertainty quantification via mean-field equations)

  Based on the expressions for the expected maximum, we compute new sets of mean field equations for general beliefs, thus generalising the work of (Stamatescu, 2022), which considered one representative mean field equation. Numerical studies are presented which reveal the accuracy of the equations, under conditions for which the mean field theory is expected to hold, and how the equations' accuracy breaks down outside of these conditions. We emphasise in particular the role of sparsity in the break down in the mean field theory and the equations' accuracy. We compare this accuracy to other estimates of the posterior means, showing the breakdown appears to be smooth. We also compare our equations to the *implicit* estimates of several Bayesian RL algorithms under the mean field settings.

- (Certainty equivalence as a lower bound)

  Notably, we show that under the mean field conditions, the *certainty equivalence* heuristic, which replaces the model parameters with their means, is in fact a lower bound to the mean of the value functions. In the literature this is often erroneously termed the "mean MDP" (Sorg et al., 2012). This simple fact follows from an application of Jensen's inequality to the mean field equations.

- (Contraction mappings and stability)

  A suite of approximate or exact mean field equations are proven to be either contraction mappings, stable, or both. We demonstrate the analytical difficulty for proving these results for the DMFP equations in general, an important and open mathematical challenge.

### 1.3 Outline of the paper

After introducing some background material on MDPs and the dynamic mean field theory (DMFT), the paper divides into three parts. The first contains an exhaustive review of formulas, approximations and bounds for the expected value of the maximum of a set of random variables. We provide a simple comparative analysis of the bounds, exact formulae and sample estimation to obtain insights into the numerical performance of each approach. The second part turns to the reinforcement learning problem. We simulate the DMFP, computing the mean of the Bellman optimality equation based on our comparative analysis of formulae for the expected maximum. We study the performance of the mean field equations for MDPs under different Dirichlet priors, including improper priors, and then consider the effect of sparsity in the transition function and in turn Dirichlet beliefs. From this we then compare the bias of existing Bayesian RL algorithms. The final part of the paper is dedicated to establishing the stability and contractive properties of the DMFP equations and their approximations. We present both positive and negative results and an open problem.

## 2 Background

We briefly discuss the foundational concepts underlying this paper. These concepts are Markov decision processes and the dynamic mean field theory. In both of these cases our interest is not to develop the general theory of these ideas, but to provide the context within which our work makes sense. We provide significantly more detail on the dynamic mean field theory than on Markov decision processes, as the former is a relatively recent development, see (Stamatescu, 2022). For more details on Markov decision processes see (Kallenberg, 2011), (Thomas, 2007) or (Bertsekas, 2022).

### 2.1 Markov decision processes and Bayesian reinforcement learning

We define a Markov decision process (MDP) as a tuple; $\mathcal{M} = (S, A, P, R, \gamma)$. The sets

$$S = \{s : s = 1, ..., N\} \quad \text{and} \quad A = \{a : a = 1, ..., K\}$$

are the *state space* and the *action space*, respectively. The mapping $P : S \times A \to \Delta(S)$ denotes the transition dynamics of the system. For each state-action pair, $P(s, a) = P_{s,a}$, is a probability distribution over states. We write

$$P_{s,a,s'} = Pr[s_{t+1} = s' : s_t = s, a_t = a]$$

for the probability of transitioning, at time $t$, from state $s$ to state $s'$ under action $a$. For our purposes we do not consider the rewards as deterministic function but instead as random variables. For example, if the rewards are Normally distributed: $r_{s,a} \sim \mathcal{N}(\rho_{s,a}, \sigma_{s,a}^2)$. The mean-rewards are bounded, as are the variances of the reward distributions. The number $\gamma \in [0, 1]$ is the discount factor.

We may use value iteration to solve our MDP, (Thomas, 2007). The equation; $V_s^t = \max_a Q_{s,a}^t$, (Bertsekas, 2012), connects the standard value functions to $Q$-value functions. The Q-value iteration equation, based on the Bellman optimality equations for the optimal $Q$-value functions, is given by

$$Q_{s,a}^{t+1} = \rho_{s,a} + \gamma \sum_{s'} P_{s,a,s'} \cdot \max_{a'} Q_{s',a'}^t. \tag{1}$$

If we consider parameter uncertainty in the distribution rewards and transition probabilities, and take a Bayesian approach, the $Q$-value functions are then random variables. We denote the mean, variance and covariance of the random $Q$-values as: $\mathbb{E}(Q_{s,a}^t) = \mu_{s,a}^t$, $\text{Var}(Q_{s,a}^t) = \nu_{s,a}^t$ and $\text{Cov}(Q_{s,a}^t, Q_{s,a'}^t) = \Sigma_{a,a'}^t$ respectively. The mean field theory, from (Stamatescu, 2022) provides a way of approximating the posterior parameters over state-action values. Under certain conditions, which we specify below, this theory tells us that the posterior statistics for the state-action values is given by the solution to a set of mean field equations. It is these equations that we refer to when we use the term *dynamic mean field programming*.

### 2.2 Dynamic mean field programming: the identical belief case

Dynamic mean field theory is an application of statistical field theory to the equations of dynamic programming, specifically in the case that the underlying MDP is uncertain and one maintains a set of Bayesian beliefs on its parameters. The general motivation behind the development of the dynamic mean field theory comes from the successful application of methods from statistical physics to topics in computer science, and specifically neural network theory. Such interaction between the fields of computer science and statistical field theory has already occurred with results in both theoretical neuroscience, see (Sompolinsky et al., 1988), and deep neural networks, see (Poole et al., 2016).

The result of (Stamatescu, 2022) is asymptotically exact under the following assumptions. Notably, the theory was derived under the identical prior setting, across all state-actions.

**Assumption 1.** (Mean-reward and transition independence assumption.) Across state action pairs, we assume independence of the mean-rewards and transition functions; $\rho_{s,a} \perp \rho_{(s,a)'}$, $P_{s,a} \perp P_{(s,a)'}$ and $\rho_{sa} \perp P_{(s,a)'}$ for all pairs $s, a$ and $(s, a)'$.

**Assumption 2.** (Flat or centred, identical Dirichlet parameters) The Dirichlet parameters are $\alpha_{s,a,s'} = c$, with $c \geq 1$, for all $s'$ and $(s,a)$ pairs. Note that with $c = 1$ we have a flat Dirichlet distribution for all state action pairs.

**Assumption 3.** (Identical mean-reward distribution) The mean-reward $\rho_{s,a}$ has an arbitrary distribution with a moment generating function that exists. We denote its mean and variance as $\mu_{\rho_{s,a}}$ and $\sigma^2_{\rho_{s,a}}$. Notably, we do not assume bounded rewards.

**Assumption 4.** (Bounded effective horizon) The discount factor $\gamma \in [0,1)$ is fixed, meaning that the effective horizon $H_{\text{eff}} = \frac{1}{1-\gamma}$ is bounded.

**Results of the DMFP theory.** With the above condition satisfied, one obtains the following consequences in the large state space limit. The first consequence is an asymptotic independence result for the $Q$-value functions. Namely, the $Q$-value functions are independent across different state-action pairs, for a given iteration $t$ (or time step, in the finite horizon case). Such a result is known as the *propagation of chaos* property Sznitman (1991).

As a consequence of these independence assumptions it is possible, when taking the expectation of an arbitrary $Q$-value function, to effectively pass the expectation over each of the terms in Equation (1), giving the following evaluation:

$$
\begin{aligned}
\mathbb{E}\big(Q_{s,a}^{t+1}\big) &= \mathbb{E}_{\rho,P,Q^t}\Big(\rho_{s,a} + \gamma \sum_{s'} P_{s,a,s'} \cdot \max_{a'} Q_{s',a'}^t\Big) \\
&= \mathbb{E}(\rho_{s,a}) + \gamma \sum_{s'} \mathbb{E}(P_{s,a,s'}) \cdot \mathbb{E}\big(\max_{a'} Q_{s',a'}^t\big) \\
&= \mu_{\rho_{s,a}} + \gamma \sum_{s'} \bar{P}_{s,a,a'} \cdot \mathbb{E}\big(\max_{a'} Q_{s',a'}^t\big).
\end{aligned}
$$

Note that the expectation does not move through the maximum function. Here, $\mu_{\rho_{s,a}}$ is the mean of the distribution of the means of the rewards, while $\bar{P}_{s,a,s'}$ is the mean over the distribution of the random variable $P_{s,a,s'}$. Additionally note that $P_{s,a}$ is the Dirichlet vector, and $P_{s,a,s'}$ is an element of that vector. The independence across vectors is needed for the factorisation of the expectation as the $Q$-value function at time $t$ is a function of all of the Dirichlet vectors, i.e., the transition probabilities.

Henceforth we write the above expectation of the $Q$-value function with the notation: $\mathbb{E}\big(Q_{s,a}^{t+1}\big) = \mu_{s,a}^{t+1}$, giving us:

$$
\mu_{s,a}^{t+1} = \mu_{\rho_{s,a}} + \gamma \sum_{s'} \bar{P}_{s,a,s'} \cdot \mathbb{E}\big(\max_{a'} Q_{s',a'}^t\big) \tag{2}
$$

which we refer to as the mean field equation.

## 2.3 Dynamic mean field programming: the general belief case

The work of (Stamatescu, 2022) derived the DMFT for the case of identical beliefs on all mean-rewards and flat Dirichlet beliefs. This implies that there is, and the derivation in fact delivers, a single mean field or DMFP equation, representative of the mean of all state-action value functions. Despite this, the DMFP equation can be straightforwardly extended to non-identical beliefs over the MDP parameters, and one obtains instead a set of $N \times K$ equations, for each mean of the Q-values.

Non-identical beliefs can arise in several ways, and it is useful to then categorise beliefs over MDP parameters in the following ways, noting that we however maintain the independence assumption between all state-action pairs. We categorise the beleifs for the mean rewards as being (i) identical or (ii) non-identical, and the Dirichlet beliefs as being (a) flat or (b) non-flat. Based on insights from mean field theory and disordered systems, one might expect that in the case that the beleifs are (a) flat, the mean-field theory will hold regardless of whether the mean-rewards are identical (i) or not (ii).

However, as the Dirichlet beliefs become non-flat, we are essentially introducing sparisity and we would not expect the mean field theory to hold. The question of exactly how sparse, or at what rate relative to the system size $N$, is generally a very difficult technical question.

Nevertheless, we expect the set of DMFP equations to be a good first order approximation. However, one must first of all be able to actually compute solutions to the equations. This motivates the following section, that is, how to actually propagate moments when the random variables over which we take the maximum are non-identical, which was not properly considered in (Stamatescu, 2022).

### 2.4  Discussion of DMFP theory

One way to visualise the case where the dynamic mean field theory holds throughout episodes of reinforcement learning, and one that will especially serve the purpose of illustrating how we utilize it within this paper, is to envision a large, highly connected, Markov decision process. By high connectivity it is meant that an agent can transition from one state to another state from a large set of potential next states.

For a Bayesian agent, the transition probabilities, rewards, and thus value functions are random variables due to Bayesian uncertainty. Under the highly connected assumption then, at any point during learning, the agent will believe they can transition to many states. Thus the influence of the *uncertainty* of any state-action value function on another is negligible, because the agent beliefs about the transitions are themselves dense. As the number of states increases this influence of the uncertainty approaches zero. In other words, an effective independence between value functions at different state-action pairs is obtained in the large state space limit.

In general MDPs will not be highly connected, but have sparse connections. Thus as an agent learns they will eventually not believe they can transition anywhere, and the mean field result will break down. For this reason, the mean field result may be thought to describe well the start of learning, though it may hold for long periods during learning, or for highly connected MDPs. Our simulation study below investigates certain sparse systems, showing that this prediction appears to hold.

We can see from the mean field equation that we should expect that the mean state-action value $\mu_{s,a}^{t+1}$ is equal to $\mu_{\rho_{s,a}} + C$ for some constant $C$. Essentially, our first objective in this work is to determine this constant precisely, so that we may obtain exact numerical values for the Q-values. This objective motivates our study of the expected value of the maximum of sets of Normally distributed random variables.

Outside of the mean field settings, the DMFP equations will not be of the form $\mu_{\rho_{s,a}} + C$, and the constant $C$ will depend on the state-action pair, in general.

*Example.* Suppose that the mean rewards are Normal, that is $\rho_{s,a} \sim \mathcal{N}(\mu_{\rho_{s,a}}, \nu_{s,a})$, then for each state-action pair, the fixed point $Q_{s,a}^* \sim \mathcal{N}(\mu_{\rho_{s,a}} + C, \nu_{s,a})$. Similarly we could take mean rewards from any other univariate distribution, such as a Beta distribution, and our fixed points would be distributed with this distribution.

As a final remark, let us consider the variance of the state-action values. The DMFP theory predicts that the variance is that of the mean rewards. However, outside of the mean field settings, we can generally expect higher variance due to stronger dependence between Q-values. The exception to this of course is if we have many samples for each state-action pair, and the parameters are essentially known.

## 3  Comparison of bounds and estimates

In this section we introduce the different methods for computing and approximating the expected value of a set of independently distributed random variables. We focus in particular on Normal random variables, though we find the best approximation formulae holds for arbitrary distributions.

Bounds for the expected maximum are perhaps what one might consider first, especially in the discipline of cmputert science. These are often justified by whether they are "tight" or not, but this refers to the number of elements considered in the maximum going to infinity. In the RL context this corresponds to the action space, which is fixed and usually small, thus tightness of bounds is not particularly relevant. Here we

investigate a range of exact and approximate methods for calculating the expected maximum and compares to various bounds, including tight ones.

Formulas for the expected maxima that have closed forms, including those involving special functions, are referred to as 'exact'. Upper and lower bounds are referred to as either 'bounds' or 'approximations'. Anything requiring numerical integration is strictly an approximation, and Monte-Carlo samples are 'estimates'

In Section (3.1) we introduce equations for the expectation of the maximum. We detail their parameter constraints, derivations, and we provide a short summary of some of the limitations we have found through simulation work and also in theoretical work. The different upper and lower bounds that we study are also introduced in Sections (3.5) to (3.8), and are given the same treatment as the exact formulae. Our only approximation method that is not a bound or exact formula is the method of Monte-Carlo sample estimation. While we ultimately settle on one of the exact formulas as our 'standard', against which we measure the rest, the Monte-Carlo sample estimations also provide a decent benchmark for determining the general degree of accuracy that each bound and exact value provides.

Lastly we summarise the results of our simulations in Section (3.9). We tabulate the numerical results of each expression for a quick evaluation of the difference in precision between them, and we generate plots for selected expressions, showing the change in the numerical values as the number of random variables is increased.

### 3.1 Exact formula for independent, non-identical random variables

For independent, non-identical random variables from an arbitrary distribution we can derive a useful formula for the expected maximum, as follows. Let $X$ be a random variable with cumulative distribution function $F(x)$; by definition

$$\mathbb{E}(X) = \int_{-\infty}^{\infty} x \, dF(x). \tag{3}$$

A well known way to write the expectation, see (Feller, 1991), is as

$$\mathbb{E}(X) = \int_{0}^{\infty} 1 - F(x) \, dx - \int_{-\infty}^{0} F(x) \, dx. \tag{4}$$

Equation (4) can then be written as

$$\mathbb{E}(X) = \int_{0}^{\infty} 1 - F(x) - F(-x) \, dx, \tag{5}$$

which can be found in a more general form in (David, 1981), which can be derived from formulas in (Tippett, 1925) and (Cox, 1954). It is then shown in (Ross, 2003) that if $X_1, ..., X_k$ are independent random variables with cdf's $F_{X_1}, ..., F_{X_k}$ respectively, then the above formula can be used to write,

$$\mathbb{E}(\max_{i} X_i) = \int_{0}^{\infty} 1 - \prod_{i=1}^{k} F_{X_i}(x) - \prod_{i=1}^{k} F_{X_i}(-x) \, dx. \tag{6}$$

In our simulations the cumulative distribution functions will be Normal c.d.f.'s written as error functions. We will sometimes also refer to this equation as the "Ross" exact formula.

### 3.2 Exact formula for multivariate Normal variables

A less well known exact formula for the expected maximum can be found in a paper by Biyi Afonja, (Afonja, 1972). For a multivariate Normal random variable of dimension $k$, we write $\mathbf{X} \sim \mathcal{N}_k(\boldsymbol{\mu}, \boldsymbol{\Sigma})$, with $k$-dimensional probability density function $f_k(\mathbf{X}, \boldsymbol{\mu}, \boldsymbol{\Sigma})$ to emphasise the mean vector $\boldsymbol{\mu}$ and covariance matrix $\boldsymbol{\Sigma}$ used in the formula for the multidimensional Normal p.d.f., (Moran, 1968). We denote the standardized p.d.f. for the standardised variable $z_i = (X_i - \mu_i)/\sigma_i$ by $f_k(\mathbf{z}; \mathbf{R})$ where $\mathbf{R}$ is the correlation matrix of $\mathbf{X}$. We write $\Sigma_{i,j}$ for the covariance between $X_i$ and $X_j$, and $\sigma_i^2 = \Sigma_{i,i}$.

For a $k$-dimensional vector $\mathbf{b} = (b_1, ..., b_k)$ we write the complementary cumulative distribution function for the standardised probability density function as

$$\bar{F}(\mathbf{b}; \mathbf{R}) = \int_{\mathbf{b}}^{\infty} f_k(\mathbf{z}; \mathbf{R}) \, d\mathbf{z} = \int_{b_1}^{\infty} \cdots \int_{b_k}^{\infty} f_k(\mathbf{z}; \mathbf{R}) \, d\mathbf{z}. \tag{7}$$

We now define the following piece-wise function:

$$\lambda_{i,j} := \begin{cases} (\mu_j - \mu_i)/\sqrt{\operatorname{var}(X_i - X_j)} & i \neq j \\ -\infty & i = j \end{cases}. \tag{8}$$

If we then define a vector $\lambda_i = (\lambda_{i,1}, ..., \lambda_{i,k})$ in $\mathbb{R}^k$, we can then define another vector in $\mathbb{R}^{k-1}$ of the form $\boldsymbol{\lambda}_i = \lambda_i$ with the $i$th element removed. For example: $\boldsymbol{\lambda}_1 = (\lambda_{1,2}, ..., \lambda_{1,k})$, $\boldsymbol{\lambda}_2 = (\lambda_{2,1}, \lambda_{2,3}, ..., \lambda_{2,k})$, and so on.

We also define another vector, this time in $\mathbb{R}^{k-2}$, as $\boldsymbol{\lambda}_{i,j} = \{\lambda_{i,jj'}\}$ for $j \neq j'$, $j, j' \neq i$, where each element is defined by the equation:

$$\lambda_{i,jj'} := \frac{\lambda_{i,j'} - \lambda_{i,j} r_{i,jj'}}{\sqrt{1 - r_{i,jj'}^2}} \tag{9}$$

with the symbol $r_{i,jj'}$ defined in accordance with the matrix $\mathbf{R}_i$ below.

We construct a new correlation matrix of correlations between differences of the random variables: $\mathbf{R}_i = \{r_{i,jj'}\}_{j,j'=1}^k$, where for $j, j' \neq i$, the symbol $r_{i,jj'}$ denoted the correlation between $X_i - X_j$ and $X_i - X_{j'}$. Similar to before, we can then define another matrix: $\mathbf{R}_i^+$ as the matrix $\mathbf{R}_i$ with the $i$th row and column removed. Lastly, we define the matrix $\mathbf{R}_{i,j} = r_{i,qs \cdot j}$, where $r_{i,qs \cdot j}$ is the partial correlation between $X_i - X_q$ and $X_i - X_s$ given $X_i - X_j$. It is shown in (Baba et al., 2004) that for multivariate Normal distributions the partial correlation is equivalent to the conditional correlation, a fact that can be useful for calculating the values of $\mathbf{R}_{i,j}$.

Letting $f_1(x)$ be the standard Normal probability density function in 1-dimension, we define a new symbol

$$\Lambda_{i,j}(x) = \frac{\sigma_i^2 - \boldsymbol{\Sigma}_{ij}}{\sqrt{\sigma_i^2 + \sigma_j^2 - 2\boldsymbol{\Sigma}_{ij}}} f_1(x) \tag{10}$$

which allows us to write the "Afonja" equation for the expected maximum of a set of Normal random variables, as given in Equation (3.1) of (Afonja, 1972) as

$$\mathbb{E}(\max_i X_i) = \sum_{i=1}^k \mu_i \bar{F}_{k-1}(\boldsymbol{\lambda}_i; \mathbf{R}_i^+) + \sum_{i=1}^k \sum_{j \neq i} \Lambda_{i,j}(\lambda_{i,j}) \bar{F}_{k-2}(\boldsymbol{\lambda}_{i,j}; \mathbf{R}_{i,j}) \tag{11}$$

where

$$\bar{F}_{k-1}(\boldsymbol{\lambda}_i; \mathbf{R}_i^+) = \int_{\lambda_{i,n_1}}^{\infty} \cdots \int_{\lambda_{i,n_{k-1}}}^{\infty} \frac{1}{\sqrt{(2\pi)^{k-1}|\mathbf{R}_i^+|}} \exp\left[ -\frac{1}{2}(\mathbf{z})^t \left[\mathbf{R}_i^+\right]^{-1} (\mathbf{z}) \right] d\mathbf{z} \tag{12}$$

$$\bar{F}_{k-2}(\boldsymbol{\lambda}_{i,j}; \mathbf{R}_{i,j}) = \int_{\lambda_{i,m_1}}^{\infty} \cdots \int_{\lambda_{i,m_{k-2}}}^{\infty} \frac{1}{\sqrt{(2\pi)^{k-2}|\mathbf{R}_{i,j}|}} \exp\left[ -\frac{1}{2}(\mathbf{z})^t \left[\mathbf{R}_{i,j}\right]^{-1} (\mathbf{z}) \right] d\mathbf{z} \tag{13}$$

$n_1, ..., n_{k-1}$ and $m_1, ..., m_{k-2}$ being the appropriate indices, according to the expansion of (11). This is why we specified the matrix included in the definitions of the probability density function and the cumulative distribution function earlier. Equation (11) is equivalent to *Equation 3.1* in (Afonja, 1972), we have simply altered the notation for succinctness.

### 3.3   Exact formula for standardised, independent and identical Normal variables

In the more restricted setting of standardised I.I.D. Normal random variables, an exact formula for the expected maximum is given by (Kamath, 2015). Suppose $X_i \sim \mathcal{N}(0,1)$ for $i = 1, ..., k$, then the equation

$$\mathbb{E}(\max_i X_i) = \int_{-\infty}^{\infty} x k f_X(x) F_X(x)^{k-1} \, dx \tag{14}$$

can be derived by taking the derivative of the product of cumulative distribution functions. $f_X(x)$ and $F_X(x)$ are the standard Normal density function and cumulative distribution function respectively. Substituting the particular expressions for a Normal distribution gives the equation

$$\mathbb{E}(\max_i X_i) = \int_{-\infty}^{\infty} \frac{xk}{2\pi} e^{-\frac{x^2}{2}} \cdot \left[ \frac{1}{2} + \frac{1}{2} \operatorname{erf} \left( \frac{x}{\sqrt{2}} \right) \right]^{k+1} \, dx. \tag{15}$$

Here, $\operatorname{erf}(x/\sqrt{2})$ is the error function evaluated at $x/\sqrt{2}$, see (Andrews, 1998). Because this expression for the expected maximum is only applicable for I.I.D. Normal variables we do not include the results of our simulations of it in this paper. However we note that during those simulations Eq. (15) agreed with both Afonja (11) and Ross (6) to at least 16 decimal places.

### 3.4   Limitations of the exact formulae

Each of the exact formulas above has its own advantages and disadvantages. The exact formula given by (Afonja, 1972) works for multivariate correlated Normal variables, that is, it places no restrictions on the parameters of the distribution. It also has many simplifications when the means of the random variables are equal. One unfortunate drawback of Equation (11) is that there seems to be no way to avoid higher dimensional numerical integration. This prevents us from effectively studying the DMFP using this formula when there are more than 3 random variables. Even with the simplifications when the means are equal, we are limited to $k \leq 5$ random variables when doing our expected maximum comparisons. As found in (Afonja, 1972), (Gupta, 1963) and (Tallis, 1961), formulas exist for computing the multiple integrals in Equation (11) for values of $k$ greater then 5, however they are largely impractical for simulation work and so we do not use them. Equation (15), as we have already stated, is restricted by its parameters, however it is much faster to compute. Equation (6) is the most useful of the three that we have found. It has broader applicability than (15), and can be applied in higher dimensions without any significant decrease in accuracy, unlike (11). These conclusions we draw from our simulations which we present in Section (4).

As previously mentioned, in our case the cumulative distribution functions are those of a Normal distribution and we can write (6) as the integral over products of error functions. The reason for the accuracy given by this formula is that error functions can be computed to very high precision numerically. Further, since we only have a single integral to compute regardless, we are not hampered by any dimensionality issues which gives us the relatively greater accuracy. Additionally, these facts hold true for any number of random variables, meaning we can use this formula to run DMFP simulations in much higher dimensions, when our action space is larger than 2. We will see the effect that this has on through our simulations later in Section 3 and in Section 4.

Before we discuss the upper and lower bounds we would like to address the question as to why, if exact formulae for the expected maximum exist, do people still consider bounds and other approximations. Other than the limitations already specified there are two broad issues that the exact formulas face. The first is that the numerical computations are not simple when using the exact formulas, unless the number of variables is very small, or the parameters are heavily restricted. Even Equation (6), which was found to be the most applicable throughout our work, requires one to integrate over products of cumulative distribution functions, which in our simulations almost always meant error functions. When compared to some of the bounds this extra computational work may be excessive. A second possibility is that the exact formulas do not lend themselves to the simple algebraic manipulations desirable in theoretical work. As we will make clear in the later sections using the exact formulas is technically more difficult than using many of the bounds.

### 3.5 A lower bound via Jensen's inequality and the certainty equivalence heuristic

Jensen's inequality provides us a straightforward lower bound on the expected maximum:

$$\mathbb{E}(\max_i X_i) \geq \max_i \mu_i. \tag{16}$$

In the context of the DMFP equations, we note that this implies that the Jensen lower bound provides the so-called "certainty equivalent" control (Duff, 2002). Explicitly, the DMFP equations under this approximation read as,

$$\mu_{s,a}^{t+1} = \mu_{\rho_{s,a}} + \gamma \sum_{s'} \bar{P}_{s,a,s'} \cdot \max_{a'} \mu_{s',a'}^{t}. \tag{17}$$

We see that this expression simply replaces the model parameters by their averages. In general in discrete state and action RL this is considered a good approximation to the mean of the value functions, which we can interpret as the certainty equivalent (CE) approximation. Originally, CE was derived for the case of linear quadratic Gaussian control, where the state variable is uncertain but the model is known (Theil, 1957), in which case it is the optimal strategy. In the case of model uncertainty it has heuristically been applied for decades, and more recently adapted to provide algorithms with learning guarantees for linear quadratic control (Mania et al., 2019).

The relationship between CE and the posterior mean, for which it is a lower bound, was not noted in the earlier work introducing DMFP (Stamatescu, 2022).

### 3.6 Upper bounds for multivariate Normal variables

The first upper bound we consider is given by (Ross, 2003):

$$\mathbb{E}(\max_i X_i) \leq c + \sum_i^k \int_c^\infty Pr[X_i \geq x]\, dx \tag{18}$$

where the optimal value of $c$, that is, the one that gives the lowest upper bound, is found by solving: $\sum_i Pr[X_i > c] = 1$. The practice of finding $c$ makes this upper bound somewhat slower and more complicated than the exact equation given in (Ross, 2003), Equation (6). We determined $c$ exactly for the 2-dimensional cases only, in all other simulations $c = 2$.

One upper bound that we study, and will be useful in a later analysis of the exact expression of the expected maximum for bivariate Normal random variables, is given in a paper by (Aven, 1985). We have for an arbitrary set of Normal random variables, "Aven's" upper bound:

$$\mathbb{E}(\max_i X_i) \leq \max_i \mu_i + \frac{1}{\sqrt{k}} \Big( (k-1) \sum_i \sigma_i^2 \Big)^{\frac{1}{2}}. \tag{19}$$

which importantly for our simulations, is true for Normal random variables under any parameter domain. We will use this bound repeatedly when drawing conclusions from the DMFP simulations and comparing it to the exact formula (6).

From (Bertsimas et al., 2006) we find the following upper bounds for the expected maxima when $\mathbf{X} \sim_{\boldsymbol{\theta}} (\boldsymbol{\mu}, \boldsymbol{\sigma}^2)$. While it is not specified in (Bertsimas et al., 2006) exactly what type of distribution $\boldsymbol{\theta}$ is, in this paper we will consider it a Normal distribution with arbitrary means and variances with all variables independent. Writing

$$\alpha_i = \mu_i + \frac{k-2}{2\sigma_i \sqrt{k-1}},$$

we can then write the two upper bounds as:

$$\mathbb{E}(\max_i X_i) \leq \frac{1}{2} \sum_i \left( \mu_i + \sqrt{(\mu_i - \max_i \alpha_i)^2 + \sigma_i^2} \right) + (2-k) \cdot \max_i \alpha_i \tag{20}$$

and

$$\mathbb{E}(\max_i X_i) \leq \frac{1}{2} \sum_i \left( \mu_i + \sqrt{(\mu_i - \min_i \alpha_i)^2 + \sigma_i^2} \right) + (2 - k) \cdot \min_i \alpha_i. \tag{21}$$

Under certain parameter conditions described below, these two upper bounds can be reduced to equation (19).

### 3.7 Tight upper bound for identical Normal variables

(Bertsimas et al., 2006) also give us a tight upper bound on the expected maxima for variables with equal means and variances, that is, when $X_i \sim_{\boldsymbol{\theta}} (\mu, \sigma^2)$ for all $i$:

$$\mathbb{E}(\max_i X_i) \leq \mu + \sigma \sqrt{k - 1} \tag{22}$$

An important point to make is that because our MDP is not asymptotic in the action space, tightness of the upper bound is not relevant.

### 3.8 Bounds for zero mean, i.i.d. Normal variables

From (Kamath, 2015) we get an upper and lower bound for $k$ many independent Normal variables $X_i \sim \mathcal{N}(0, \sigma^2)$:

$$\mathbb{E}(\max_i X_i) \leq \sigma \sqrt{\log k^2} \tag{23}$$

and

$$\mathbb{E}(\max_i X_i) \geq \frac{1}{\sqrt{\pi \log 2}} \sigma \sqrt{\log k}. \tag{24}$$

While the parameter domain of these bounds are limited to zero mean and equal variances, we will find in our analysis that the lower bound performs much better, both to Equation (23) and to some of the other upper bounds.

Since the different bounds above have different parameter constraints we compute numerical values for each of them over specific parameter domains on which they are well defined. Numerical results for all allowed parameter domains under consideration are given for completeness, and for a more comprehensive comparison of the bounds, approximations and exact values.

### 3.9 Numerical simulation results

The Monte-Carlo sample estimation results are given in the table below for $10^8$ samples of maxima of $k$-many random variables. It is important to note that the Monte-Carlo sample estimation has a very slow rate of convergence to the true value, and so a high number of samples is required even for the generally weak agreement we see between the Monte-Carlo and exact values in the tabulated data below.

As mentioned earlier, there are certain parameter limitations for which the various exact values and bounds are defined. For this reason we compute numerical values for the formulas across a range of different parameter conditions. Again, throughout this paper we are interested in the DMFP for Normally distributed random variables, and this section is no different. Due to a desire to obtain as many numerical results for the exact formula given by Afonja (11), we generated tables for $k = 2, 3, 4$ and $5$ many random variables. All of the random variables come from a $k$-dimensional Normal distribution, in each column our means and variances, when not 0 or 1, were randomly generated from a Uniform$(0, 1)$ distribution, and kept the same for each equation in that column.

|  | $\mathcal{N}_5(0, \sigma^2)$ | $\mathcal{N}_5(\mu, \sigma^2)$ | $\mathcal{N}_5(\mu, \sigma_i^2)$ | $\mathcal{N}_5(\mu_i, \sigma_i^2)$ |
|---|---|---|---|---|
| Monte-Carlo | 1.1261417367 | 1.4278245252 | 1.9965215452 | 1.4667868827 |
| (6) Ross | 1.1261270919 | 1.4278005320 | 1.9965940928 | 1.4668164213 |
| (11) Afonja | 1.1261270919 | 1.4278005320 | 1.9965940928 | - |
| (18) Ross upper | 1.1515498802 | 1.4885300836 | 2.6062061621 | 2.0133558752 |
| (19) Aven | 1.9366491711 | 1.7477860941 | 2.7272233051 | 2.5949895751 |
| (20) Bertsimas MAX | 1.9366491711 | 1.7477860941 | 2.7231229198 | 2.204443024 |
| (21) Bertsimas MIN | 1.9366491711 | 1.7477860941 | 2.7244871754 | 2.3534389117 |
| (22) Bertsimas T | 1.9366491711 | 1.7477860941 | - | - |
| (23) (Kamath upper) | 1.7372930018 | - | - | - |
| (24) (Kamath lower) | 0.8324734769 | - | - | - |

Table 1: *Numerical values for $\mathbb{E}(\max_i X_i)$ using exact values, bounds and Monte-Carlo estimation under different parameters constraints on the Normally distribution random variables. $k = 5$*

Aside from the convergence of the Monte-Carlo sample estimation, there are a few key features that can be observed from the information given in the table above. The first point of interest is the equivalent numerical values given by Equations (19), (20), (21), and (22) under the parameter condition of all random variables having equal means and variances, as alluded to in the previous subsection. The second point is the accuracy of the upper bound given in (Ross, 2003), Equation (18).

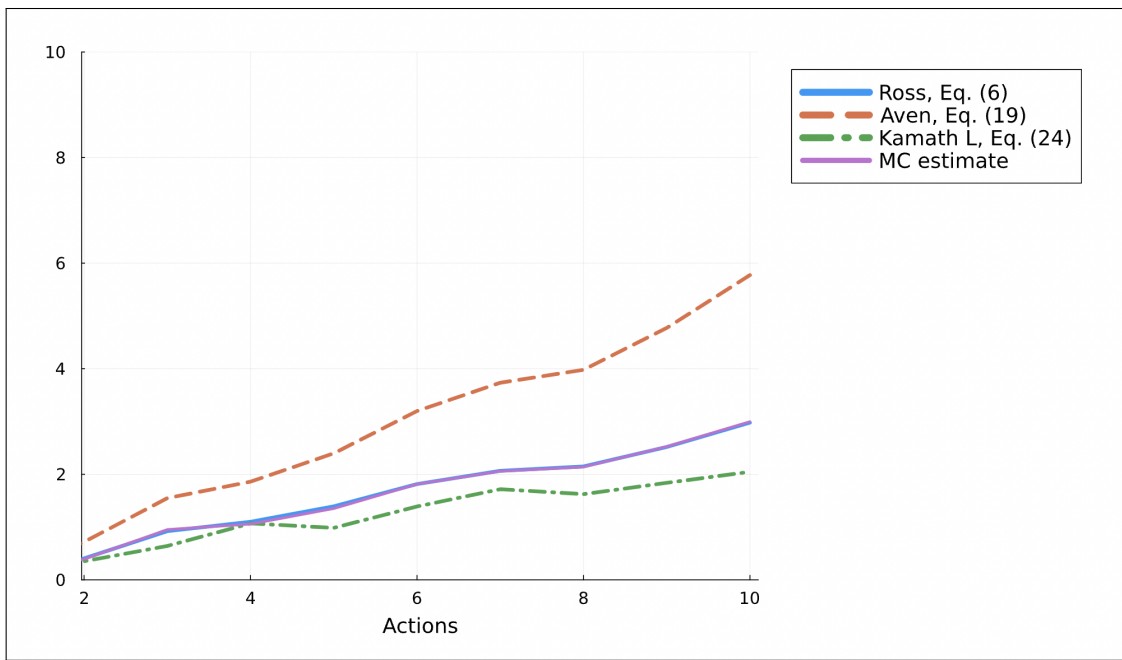

Figure 2: *Numerical values of eq. (6), eq. (19) and eq. (24). We use Normal distributions, $\mathcal{N}_k(0, \sigma_i^2)$ $k = 2, 3, ..., 10$. We also plot the Monte Carlo estimate of the expected maximum from 2000 samples, which agrees with the estimate of (6.*

In Figure 3 we visualise some of the results above for higher dimensional Normal random variables. Due to the relatively similar performance of the expressions (19), (20), (21) and (22) we only simulate (19) among these bounds. Of the three exact formulas for the expected maximum we only simulate Equation (6) as it

can be extended to higher dimensional spaces and holds for arbitrary means and variances. Lastly we have added the lower bound given by (Kamath, 2015), and a spread of samples obtained via Monte-Carlo sample estimation, represented by a series of points distributed above the number of random variables in the Normal distribution.

Only 20 samples are used to obtain our Monte-Carlo estimates as anything above this reduces the spread in the empirical distribution to the point of collapsing them onto one another.

## 4 DMFP simulation study

In this section we present simulations of the approximate DMFP mean field equations, based upon equation (2) and obtained using equations (6), (11), (19), (20) and (21) respectively. The simulations are organised in two parts.

First, we demonstrate that under the conditions where we expect the mean field theory to hold (detailed in Section 4), the DMFP mean field equation accurately computes the mean of the distribution of fixed points. We demonstrate this by comparing the Monte Carlo estimates of the mean of the optimal Q-values to the DMFP estimate, based on the various approximations. The calculations based on numerical integration, put forward by Ross, performs best.

Second, we study the DMFP equations outside of the mean field setting. We focus on two cases, both of which increase the level of sparsity in the beliefs over the transition probabilities of the MDP. Specifically, we first demonstrate that as the prior becomes improper, the mean field equations lose accuracy. Secondly, we show that if a particular MDP has sparse transitions (eg. only a few states are accesible from any one state), and we have interacted with this system and our beliefs become sparse, then the equations lose accuracy. The promising result is that as the system becomes sparse, the mean field theory breaks down gradually, in what appears to be a smooth manner.

### 4.1 Empirical evaluation of new DMFP equations

We first compare several DMFP approximations based on the best performing approximations in Section 3. We compare the methods by measuring the average absolute difference between the empirical Q-value mean, derived from Monte Carlo estimates, and the mean derived from the approximate DMFP equation. Tabulated below is the average absolute difference corresponding to the formula used in place of the expected maximum in Algorithm 1. We restrict ourselves to only $k = 2$ actions, since this allows use to use a closed form exact value for the expected maximum. We define

$$\textbf{Avg absolute difference } = \left| \mu_{s,a}^* - \frac{1}{500} \sum Q_{s,a}^* \right|.$$

as the average absolute difference, and derive the following:

| Equation used in DMFP | Avg absolute difference |
|---|---|
| eq. (6) 'Ross' | 0.041109405623055206 |
| eq. (11) 'Afonja' | 0.04110940562305454 |
| eq. (19) 'Aven' | 1.057967634414033 |
| eq. (20) 'Bert MAX' | 1.057967634414033 |
| eq. (21) 'Bert MIN' | 1.057967634414033 |

Table 3: $k = 2$. *Average absolute differences between posterior means computed using the DMFP and Monte-Carlo sample estimation.* 500 *samples, using the same IID Normal parameters for each DMFP expression, and using* $\mathcal{N}_2(0, 1)$ *distributed variables.*

While it is clear from the numerical results that Equation (6) provides the most accurate results, we point out that when compared in the previous analysis, both the exact formula from (Afonja, 1972), Equation (11) and the exact formula of Equation (6) gave almost identical results. This tells us that the discrepancy in outcome between the two versions of the mean field equation is a result of the code that we used in simulation, and more specifically, the numerical methods employed in that code.

It is important to note that when running the DMFP simulations and computing the posterior mean via Equation (2), we have multiple sources of error accumulation to consider. The first is that the numerical integration required to evaluate the expected maximum using Ross (6) is not truly exact, and that any numerical discrepancy will accumulate throughout the iterations. Additionally we have observed certain limitations with the method of numerical integration used in our simulations. All of our simulations have been run using the *Julia programming language*, and our numerical integration is performed by using the QuadGK.jl package. This package uses the Gauss-Kronrod quadrature method of numerical integration, see (Calvetti et al., 2000). While this method is very accurate and we have attuned its parameters for the best possible evaluations, we have also found that around the point of inflection of error functions, if the tangent becomes too steep, there is a noticeable break down in the accuracy of the numerical integration as the number of actions increases. The decrease in accuracy is not very strong however, and is consistent with the slow convergence of the expected maximum to the Gumbel distribution. Coupled with the accumulation of error that the iterations entail, and this could contribute to significant differences between the DMFP derived posterior mean and the Monte-Carlo sample estimation of the posterior mean.

We also present how well (6) performs in comparison to some of the bounds in larger actions spaces. In Figure 6 we show the overlay of the Normal distribution defined using the parameters $\mu^*_{s,a}$ and $\nu^*_{s,a}$ from the mean field equations, placed on top of the empirical distribution consisting of the samples of $Q^*_{s,a}$. We know that $\nu^*_{s,a}$ is unchanged, and $\mu^*_{s,a}$ is derived using Algorithm 1 and Equation (6), as well as Equation (19).

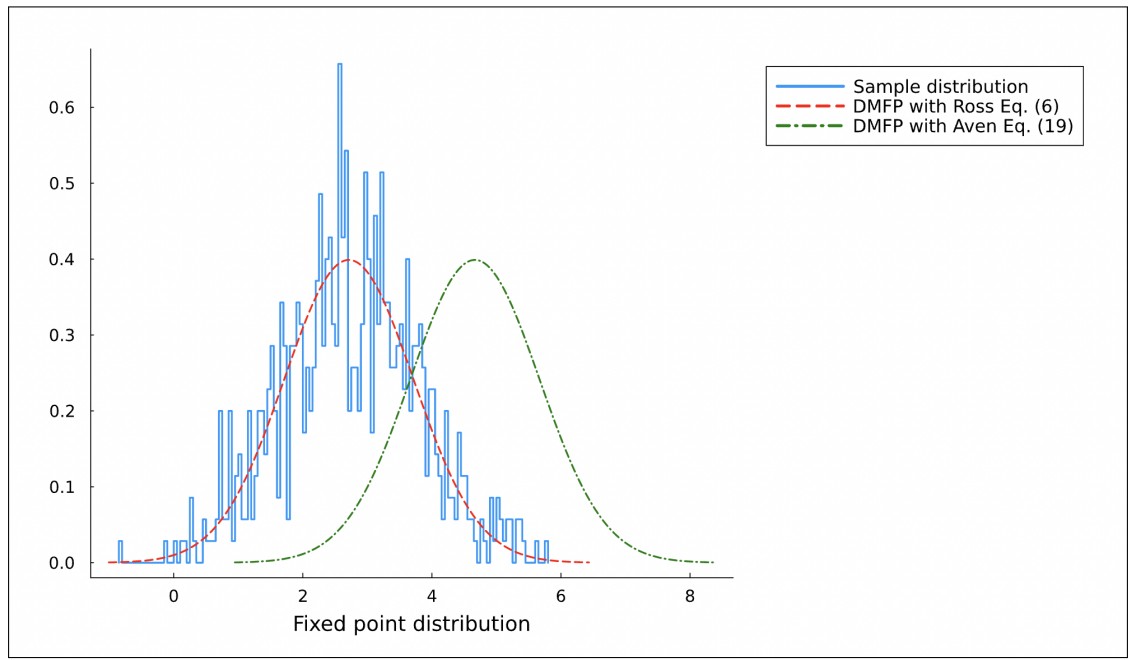

Figure 4: *Overlay of two Normal distributions with parameters derived from DMFP approximations, using eq. (6) and eq. (19) respectively, onto the sample distribution of the fixed points $Q^*$ for a single state-action pair. Our MDP had 200 states, 5 actions, a discount factor of 0.7, and we ran value iteration for 30 time-steps. We took 700 samples of $Q^*$.*

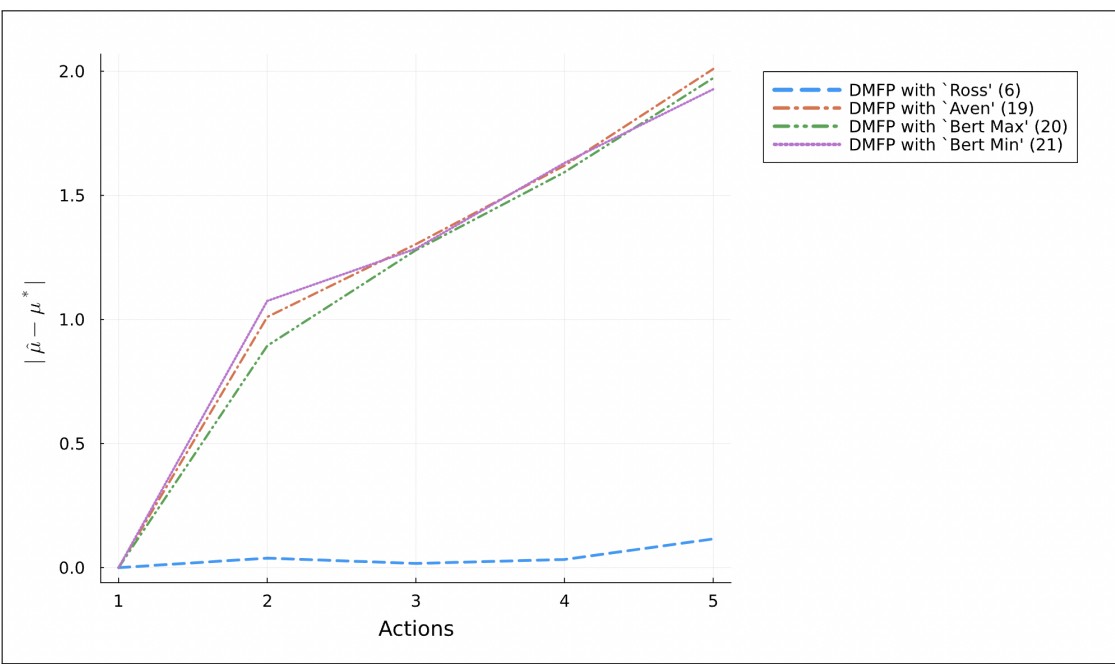

Figure 5: *Numerical values of the absolute difference between $\mu^*$, as calculated by DMFP, and the Monte Carlo estimate denoted $\hat{\mu}$, for a particular state-action pair and particular iteration, as the number of actions (and hence random variables in our expected maxima calculations) increases. The values of $\mu^*$ were computed using equations (6), (19), (20) and (21) respectively.*

We also provide a comparison plot of the accuracy of the DMFP derived posterior mean using a suite of different formulas for the expected maximum. We use Ross (6), Aven (19), and two bounds from (Bertsimas et al., 2006), 'Bertsimas MAX' (20) and 'Bertsimas MIN' (21). We add the lines $y = x$ and $y = \ln(x)$ to provide an idea of the inaccuracy of the bounds as the number of actions increases.

We can see from the results of the simulations above that the 'exact' DMFP using (6) in place of the expected maximum, computes the posterior mean almost perfectly. Even with the qualifications on the actual and potential sources of error that we mentioned at the beginning of this section. This is achieved by iterating the mean field equation, Equation (2), a total of 30 times for each formula. By comparison, at least 500 samples for Monte-Carlo estimation were required to achieve a small difference between posterior mean values. Additionally, as we have seen with the Monte-Carlo convergence rate to the exact value of the DMFP, it would require a much larger number of samples than 500, to see significant agreement between the two posterior means, proving the utility of the DMFP.

## 4.2 DMFP performance outside of mean field conditions

We now present plots demonstrating the decline in accuracy that occurs as the transition sparsity assumed in our beliefs over the underlying MDP increases. There are two ways in which we model this. The first is by varying the Dirichlet parameters $\alpha$ directly. We set $\alpha_{s,a,s'} = c$, letting $c$ range from a larger positive value, towards zero. A large value of $c$ means the Dirichlet is peaked in the centre of the simplex, meaning the agent believes they can transition anywhere (essentially they believe they are in a contextual bandit). The smaller values of $c$ correspond to having something closer to an improper prior, meaning the agent believes the transitions are sparse.

The second way we model beliefs about sparsity is to create a sparse MDP, with transitions $P_{s,a}$ sampled from a Dirichlet with some $\alpha_{\text{true}}$, and then subsequently sample from the multinomial distribution with parameters $P_{s,a}$. We consider a small number of multinomial samples, from increasingly sparse $\alpha_{\text{true}}$.

In either case, we see that the DMFP equation accuracy breaks down in a gradual way. As a way of comparison we also present the certainty equivalent heuristic, or Jensen lower bound. We present this as well, since in some literature, for example (Sorg et al., 2012), this is referred to as the "mean MDP". Clearly, this does not produce estimates of the mean state-action values, and is thus a misleading term.

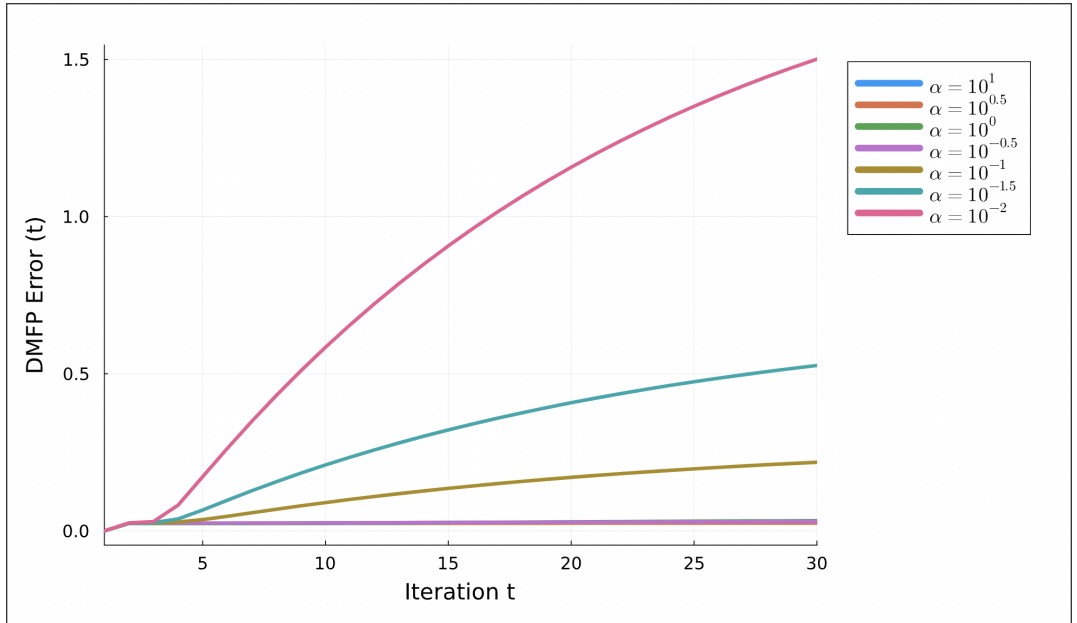

Figure 6: *Plot of the errors between the Monte-Carlo sample estimate of the DMFP, and the exact DMFP along the time-steps of a single state-action pair, for differing values of the Dirichlet parameter $\boldsymbol{\alpha} = \alpha \cdot (1, ..., 1)$. The simulated MDP has 200 states, 2 actions, a discount factor of 0.95, mean rewards with $\mathcal{N}(0, 1)$ distribution, and we take 1000 Monte-Carlo samples.*

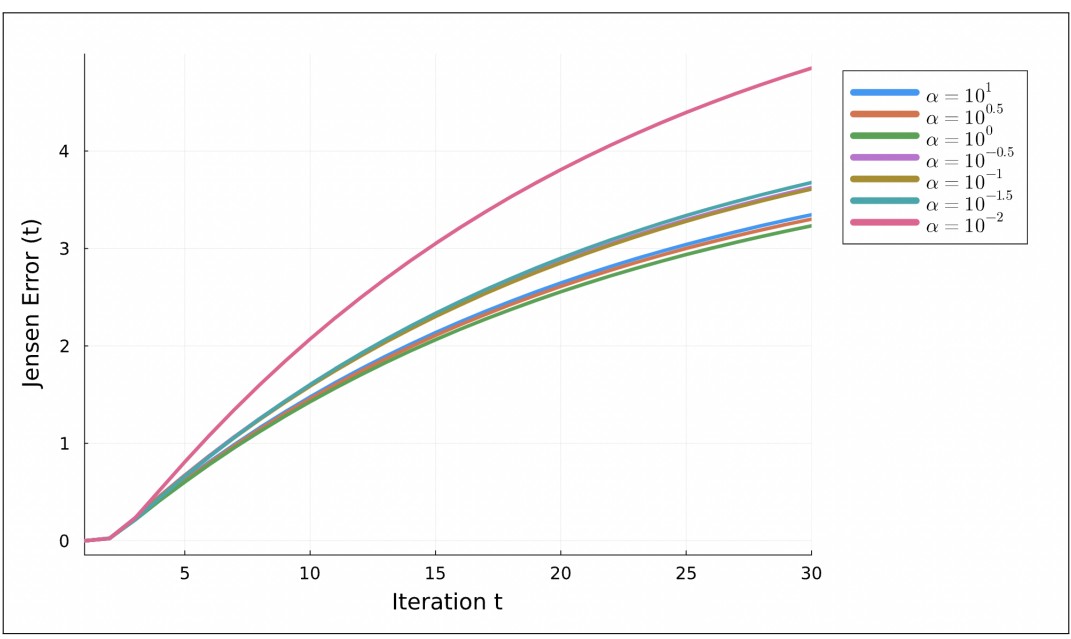

Figure 7: *Plot of the errors between the Monte-Carlo sample estimate of the DMFP, and the exact DMFP along the time-steps of a single state-action pair, for differing values of our true Dirichlet parameter $\boldsymbol{\alpha}_{true} = \alpha_{true} \cdot (1, ..., 1)$. The simulated MDP has $200$ states, $2$ actions, a discount factor of $0.95$, mean rewards with $\mathcal{N}(0, 1)$ distribution, and we take $1000$ Monte-Carlo samples.*

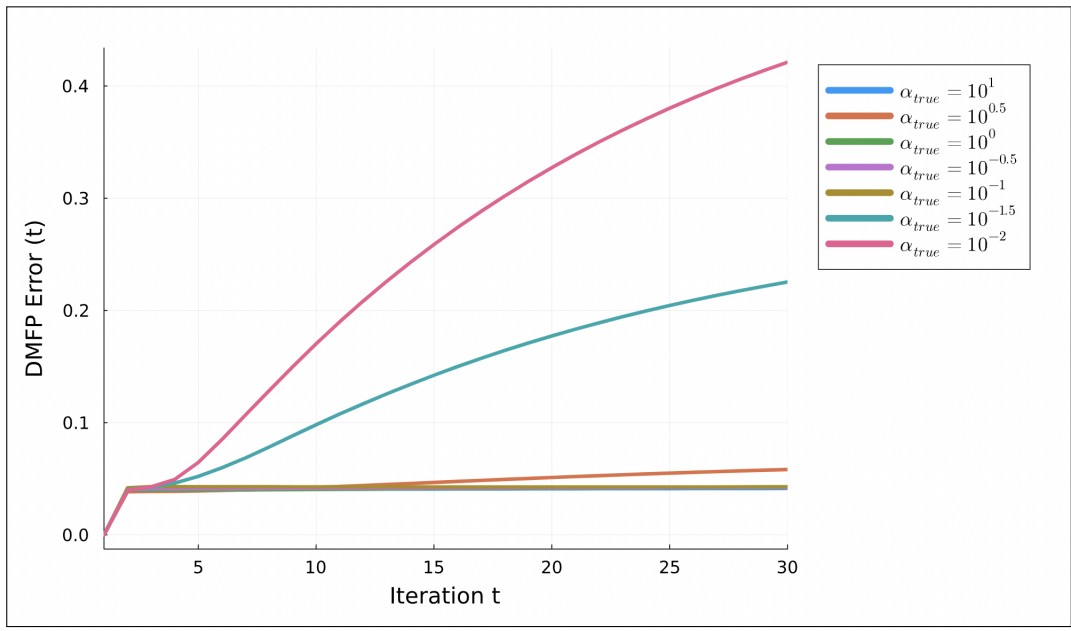

Figure 8: *Here we plot the errors between the Monte-Carlo sample estimate of the DMFP, and the certainty equivalent DMFP using Jensens inequality, for differing values of our Dirichlet parameter $\boldsymbol{\alpha} = \alpha \cdot (1, ..., 1)$. Our MDP has $200$ states, $2$ actions, a discount factor of $0.95$, our mean reward are $\mathcal{N}(0, 1)$ distributed, and we take $400$ Monte-Carlo samples.*

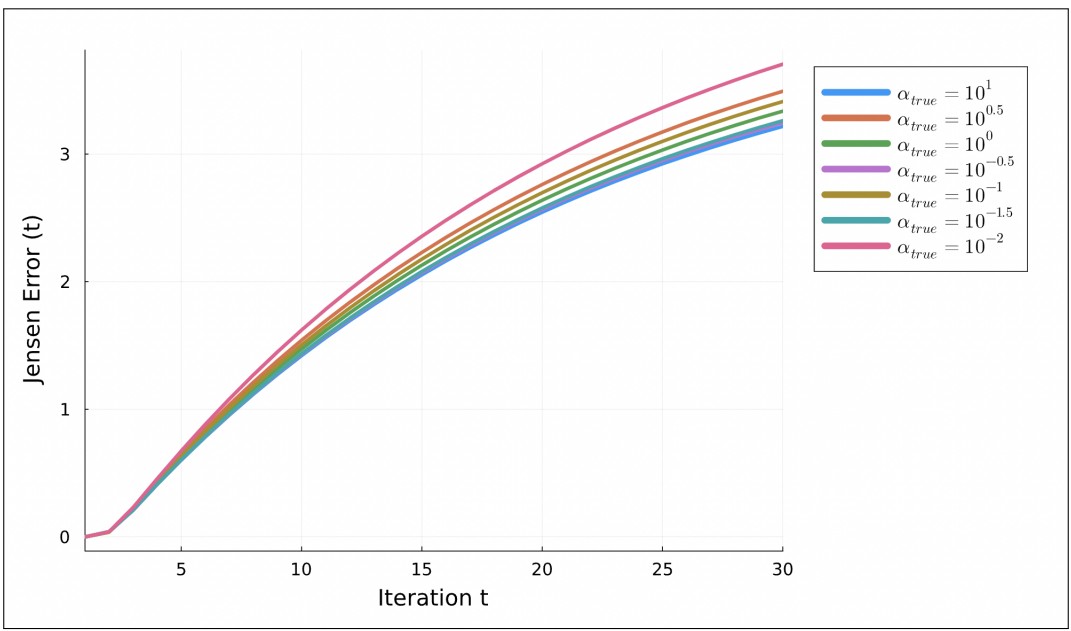

Figure 9: *Here we plot the errors between the Monte-Carlo sample estimate of the DMFP, and the certainty equivalent DMFP using Jensens inequality, along the time-steps of a single state-action pair, for differing values of our true Dirichlet parameter $\boldsymbol{\alpha}_{true} = \alpha_{true} \cdot (1, ..., 1)$. Our MDP has 200 states, 2 actions, a discount factor of 0.95, our mean rewards are $\mathcal{N}(0, 1)$ distributed, and we take 400 Monte-Carlo samples.*

As we can see, the DMFP accuracy at estimating the prior or posterior means remains quite accurate, in particular in comparison to the certainty equivalent estimate. This is promising for the utility of the theory since most MDPs in the RL literature have sparse transitions. We can report that, as one would expect, the variance of the state-action values increases as sparsity increases. This is due to the increasing dependence between the state-action values. A next-to-leading-order correction to the mean field theory would attempt to estimate this increase in the variance.

## 5   RL algorithm comparison

We now wish to provide some more direct insight into the role that the DMFP can provide in the development of reinforcement learning algorithms. We consider this in the case of a flat Dirichlet prior, where the mean field theory is exact. Under this setting, the theory predicts that beyond the first moment, the higher moments of the optimal Q-values are those of the mean rewards. This result enables us to provide some insight into the accuracy of other RL algorithms' exploration bonuses, which are often implicit.

Specifically, we consider the RL algorithms: Bayesian Exploration Bonus (BEB), (Kolter and Ng, 2009) and Variance Based Reward Bonus (VBRB) (Sorg et al., 2012). We also consider a more recently developed algorithm, K-learning (O'Donoghue, 2021). This algorithm is closer in some ways to the DMFP approach, as it explicitly considers that the Q-values are random, and tries to propagate this through a risk sensitive version of the Bellman equation.

The idea of our comparison is that the algorithms have essentially taken the certainty equivalent (or Jensen lower bound) approach, that is, plugged into the Bellman equation the mean estimates of the MDP parameters, and then added an exploration bonus for each state-action pair. As discussed in the introduction, this is the general idea of forming an index policy in the bandit approach. As such, we can infer from the certainty equivalent solution (which Sorg et al. (2012) refer to as the "mean MDP"), how large the exploration bonus is. Furthermore, given the mean field setting, we can compare this bonus to the true mean-reward standard deviation, providing valuable insight.

### 5.1 BEB algorithm

The *Bayesian exploration bonus* algorithm (Kolter and Ng, 2009), designed for the finite horizon setting, chooses actions greedily, following the $\hat{Q}$-value function,

$$\hat{Q}^t_{s,a} = \mu_{\rho_{s,a}} + \frac{\beta}{1 + n(s,a)} + \sum_{s'} \bar{P}_{s,a,s'} \cdot \max_{a'} \hat{Q}^{t-1}_{s',a'} \tag{25}$$

where $n(s,a)$ is the count for the number of times the agent has been in state $s$ and taken action $a$. Note that the same bonus type can be added when the reward is unknown, as discussed in Kolter and Ng (2009). It is this setting we compare DMFP to.

The results of our simulations presented in Figure 10 demonstrate the slight bias introduced through the modified value function in BEB. When compared to the DMFP equation for the posterior mean we can see that this bias is relatively significant, especially in the large system limit, where the mean field equation approaches perfect numerical accuracy.

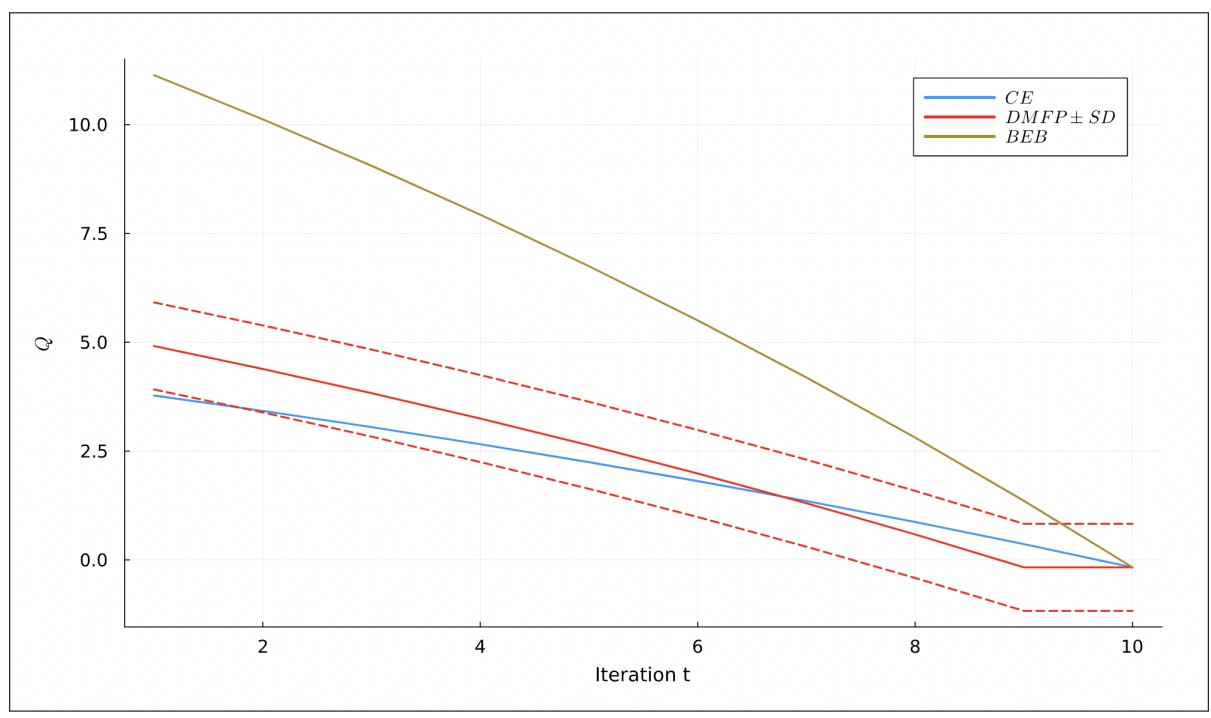

Figure 10: *Convergence of the BEB optimistic Q-value function (25), the certainty equivalence lower bound (17), and the DMFP mean field equation (2) $\pm$ the standard deviation ($= 1$). Our MDP has 200 states, 2 actions, with a discount factor of 0.7. We use $\beta = 2(20)^2$.*

In Figure 10, our comparison of the DMFP equation, and the BEB Equation (25), with the reward variance plotted around the DMFP estimate with dotted lines. We also include the certainty equivalent or Jensen lower bound estimate. We see the BEB optimistic Q-value is overly optimistic.

### 5.2 Variance based reward bonus (VBRB) and K-Learning

The *variance based reward bonus* algorithm, (Sorg et al., 2012), has an optimistic value function defined via

$$\hat{V}^*_t(s) = \max_{a \in A} \left\{ R(s,a) + \hat{R}_{s,a} + \sum_{s' \in S} P(s' \mid s,a) \hat{V}^*_{t-1}(s',) \right\} \tag{26}$$

where the exploration bonus is given by

$$\hat{R}_{s,\phi,a} = \beta_R \, \sigma_{R(s,\phi,a)} + \beta_P \sqrt{\sum_{s' \in S} \sigma^2_{P(s'|s,\phi,a)}}. \tag{27}$$

The constants $\beta_R$ and $\beta_P$ control the magnitude of the exploration bonus, while the variances in Equation (27) are defined respectively as:

$$\sigma^2_{R(s,\phi,a)} = \int_\theta R(s,\theta,a)^2 \, b(\theta) \, d\theta - R(s,\phi,a)^2, \tag{28}$$

$$\sigma^2_{P(s'|s,\phi,a)} = \int_\theta P(s' \mid s,\theta,a)^2 \, b(\theta) \, d\theta - P(s' \mid s,\phi,a)^2. \tag{29}$$

which are the variances of the rewards and Dirichlet transition probabilities respectively. Of course the optimistic $Q$-value function is

$$\hat{Q}^t(s,a) = R(s,a) + \hat{R}_{s,a} + \sum_{s'} P(s' \mid s,a) \cdot \max_{a'} \left( \hat{Q}^{t-1}(s',a') \right), \tag{30}$$

For our comparisons we use values for the parameters $\beta_R$ and $\beta_P$ taken from the respective paper. From, (Sorg et al., 2012), we set $\beta_R = \frac{1}{\sqrt{p}}$ and $\beta_P = \frac{\gamma N}{1-\gamma} \frac{1}{\sqrt{p}}$ with the probability $p = 0.9$.

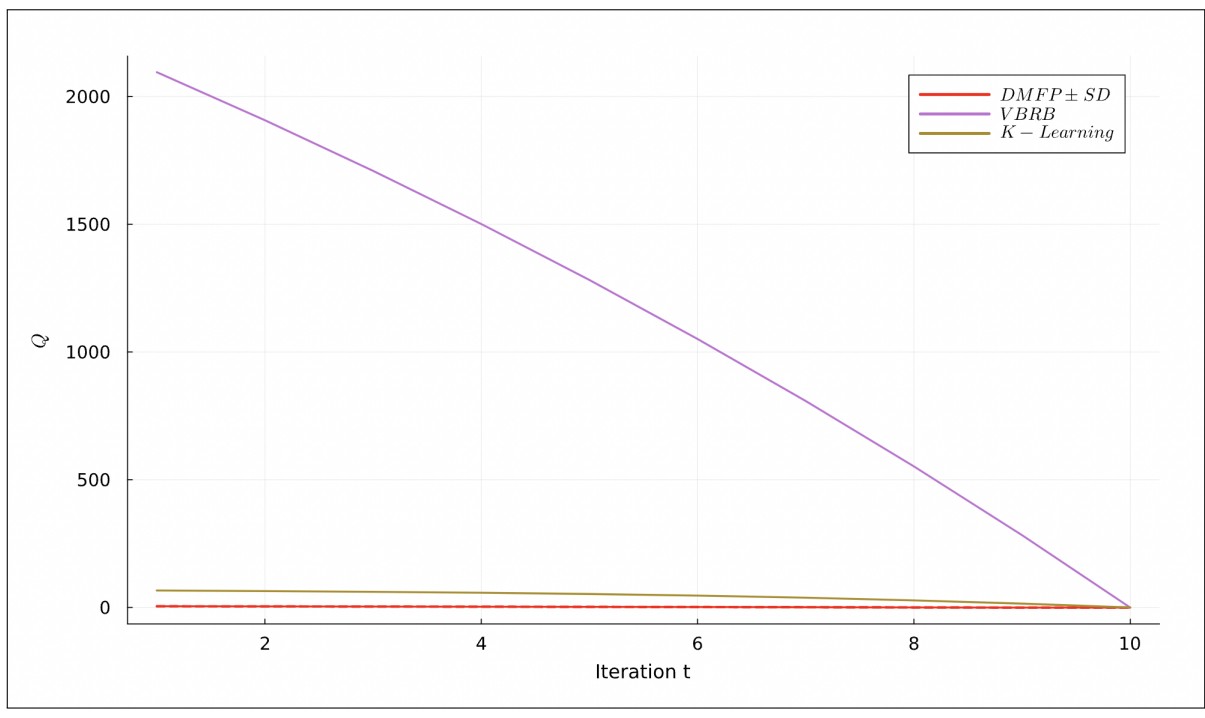

Figure 11: *Convergence of the DMFP mean field equation (2) $\pm$ the standard deviation ($= 1$), the VBRB algorithm value function (30) and the K-Learning algorithm value function (32). Our MDP has $200$ states, $2$ actions, a discount factor of $0.7$. We use $p = 0.9$ and $\tau = 3$.*

In Figure (9) above we compare the Variance based reward bonus equation for a finite horizon, to the DMFP equation, and we include the 'K-learning' Bellman optimality equation given by (O'Donoghue, 2021) for a single episode. Specifically we are interested in comparing the DMFP mean field equation to the *optimistic Bellman operator* (Eqn. (6) of (O'Donoghue, 2021)). This Bellman operator is defined as

$$\mathcal{B}_l^t(\tau,y)(s,a) = \mu^t_{\rho^l_{s,a}} + \frac{\sigma^2 + (L-l)^2}{2\tau(n^t_l(s,a) \vee 1)} + \sum_{s' \in S_{l+1}} \bar{P}^l_{s,a,s'} \left( \tau \log \sum_{a'} \exp\left[ y(s',a')/\tau \right] \right) \tag{31}$$

where $\tau \geqslant 0$, $y \in \mathbb{R}^{|S_l| \times A}$, where $n_i^t(s,a)$ is the visitation count of the learning agent to state-action $(s,a)$ at time-step $l$ before episode $t$, and the function $(\cdot \vee 1) := \max\{\cdot, 1\}$.

As we only consider a single episode, and an infinite horizon setting, we modify Equation (31) to be able to properly compare it to DMFP and VBRB.

$$\mathcal{B}^{t+1}(\tau, y)(s,a) = \mu_{\rho_{s,a}} + \frac{\nu_{s,a}^2 + (L-l)^2}{2\tau} + \gamma \sum_{s'} \bar{P}_{s,a,s'} \big( \tau \log \sum_{a'} \exp \left[ y(s',a')/\tau \right] \big) \tag{32}$$

Note that Equation (10) from (O'Donoghue, 2021) provides a formula for the "temperature" parameter $\tau$ that gives a bound on the posterior expected maximum. With the parameters of the MDP that we establish for our simulation inf Figure 9, this formula comes out to approximately 370, well above the value $\tau = 3$ in our simulation, from which we see already that the function (32) upper bounds both the DMFP equation and the VBRB equation.

### 5.3 Additional Bayesian RL algorithms

There are many other deterministic Bayesian reinforcement learning algorithms that work by altering the value function in some way, with the goal of improving some learning outcome. See (Ghavamzadeh et al., 2016) for a large collection of Bayesian reinforcement learning algorithms. Some of the algorithms seem more obviously amenable to treatment with use of the DMFT than others. Of course, there exist non-deterministic Bayesian RL algorithms. Most prominent are those based on Thompson sampling (Osband and Van Roy, 2017), but other approaches exist as well, such as the *Best Of Sampled Set* (BOSS) algorithm (Asmuth et al., 2012).

## 6 Stability analysis

Here we consider the DMFP iterations from a dynamical systems perspective and study the stability and contractive properties. As we will see the point around which we expect stability is the fixed point $\mu_{s,a}^*$, which we do not prove exists, but can observe empirically through our simulations. Throughout this section we will simply assume that the mean field equation has a fixed point, $\mu_{s,a}^*$, and that this fixed point is arrived at through iterations in exactly the same way as when finding the fixed point of the standard Bellman equation.

Formally we examine the stability of the equilibrium solution to the dynamical system corresponding to the function iteration of $\mu_{s,a}^{t+1}$. As we have considered different bounds on the expected maximum in our DMFP simulations we consider the stability of the mean field equation when the expected max is replaced by one of these formulas, specifically we look at the exact formula given in (Afonja, 1972) when the number of actions is $k = 2$, the upper bound approximation to the expected maximum from (Aven, 1985), Equation (19). We also consider the stability of the mean field equation in the large action space limit, where our expected maximum becomes a parameter for a Gumbel distribution, (Gumbel, 1954).

To show that a system is stable at a fixed point we have to compute the Jacobian matrix of the system, and determine the corresponding eigenvalues of the Jacobian. If they have absolute values less than 1, then the system is stable at that fixed point. It is clear that our fixed point is the optimal value, obtained via the iteration process on the mean field equation.

### 6.1 Avens upper bound (Aven, 1985)

**Proposition 1.** If the expected maximum term of the mean field equation (2) is substituted by the upper bound (19), then the resulting iterative function is stable around an equilibrium point is stable provided the ratio $\gamma/N$ is strictly less than 1.

*Proof.* Consider the DMFP equation with the upper bound given by (Aven, 1985) in place of the expected value of the maximum:

$$\mu_{s,a}^{t+1} = \mu_{\rho_{s,a}} + \gamma \sum_{s'} \bar{P}_{s'|s,a} \cdot \Big( \max_{a'} \mu_{s',a'}^{t} + \sqrt{\frac{k-1}{k} \sum_{a'} (\sigma_{s',a'}^{2})^{t}} \Big), \tag{33}$$

where $k$ is the number of actions/random variables. In order to show stability we compute eigenvalues of the Jacobian and show that they are strictly less than 1. We compute an arbitrary element of the Jacobian matrix, taking derivatives as follows:

$$\frac{\partial \mu_{s,a}^{t+1}}{\partial \mu_{(s,a)'}^{t}} = \frac{\partial}{\partial \mu_{(s,a)'}^{t}} \Big[ \mu_{\rho_{s,a}} + \gamma \sum_{s'} \bar{P}_{s'|s,a} \cdot \Big( \max_{a'} \mu_{s',a'}^{t} + \sqrt{\frac{k-1}{k} \sum_{a'} (\sigma_{s',a'}^{2})^{t}} \Big) \Big] \tag{34}$$

Here we have taken the partial derivative with respect to the previous time mean, at an arbitrary state-action pair.

$$\frac{\partial \mu_{s,a}^{t+1}}{\partial \mu_{(s,a)'}^{t}} = \frac{\gamma}{N} \sum_{s'} \frac{\partial}{\partial \mu_{(s,a)'}^{t}} \Big( \max_{a'} \mu_{s',a'}^{t} + \sqrt{\frac{k-1}{k} \sum_{a'} (\sigma_{s',a'}^{2})^{t}} \Big) \tag{35}$$

$$= \frac{\gamma}{N} \sum_{s'} \frac{\partial}{\partial \mu_{(s,a)'}^{t}} \Big( \max_{a'} \mu_{s',a'}^{t} \Big) \tag{36}$$

$$= \frac{\gamma}{N} \delta_{\mu_{(s,a)'}^{t}, \max_{a'} \mu_{s',a'}^{t}} \tag{37}$$

where $\delta_{\mu_{(s,a)'}^{t}, \max_{a'} \mu_{s',a'}^{t}}$ is the Kronecker delta function, defined here as

$$\delta_{\mu_{(s,a)'}^{t}, \max_{a'} \mu_{s',a'}^{t}} = \begin{cases} 1 & \max_{a'} \mu_{s',a'}^{t} = \mu_{(s,a)'}^{t} \\ 0 & \text{otherwise.} \end{cases} \tag{38}$$

We again write the Jacobian of the system using a different indexing system, for greater clarity. The Jacobian is

$$\mathbf{J} = \begin{bmatrix} \frac{\partial \mu_{1}^{t+1}}{\partial \mu_{1}^{t}} & \cdots & \frac{\partial \mu_{1}^{t+1}}{\partial \mu_{|S| \times |A|}^{t}} \\ \vdots & \ddots & \vdots \\ \frac{\partial \mu_{|S| \times |A|}^{t+1}}{\partial \mu_{1}^{t}} & \cdots & \frac{\partial \mu_{|S| \times |A|}^{t+1}}{\partial \mu_{|S| \times |A|}^{t}} \end{bmatrix} \tag{39}$$

and then once again becomes a matrix consisting of a single column of $\gamma/N$ with every other entry 0. To see this, suppose that $\max_{a'} \mu_{s',a'}^{t} = \mu_{k}^{t}$ for some $k$. Then the indicator function will return 1, only when the derivative is taken with respect to $\mu_{k}^{t}$, and it will do this for $\mu_{i}^{t+1}$ for all $i$. This is true also at the fixed point $\mu_{t}^{*}$, the point around which we expect stability. Writing out the Jacobian in terms of basis vectors we will have

$$\mathbf{J} = \frac{\gamma}{N} \mathbf{e}_{k} + 0 \cdot \sum_{j \neq k} \mathbf{e}_{j}. \tag{40}$$

Taking the determinant of $\mathbf{J} - \lambda \mathbf{I}$ to find the eigenvalues, will always give us $\gamma/N - \lambda$ times the determinant of a minor matrix equal to $-\mathbf{I}$. This gives

$$\det(\mathbf{J} - \lambda \mathbf{I}) = (\gamma/N - \lambda)(-\lambda)^{|S| \times |A| - 1}$$

and subsequently, eigenvalues of $\lambda_{1} = \gamma/N$ with all others equal to 0. As $\gamma$ and $N$ are always positive but less than 1, the eigenvalues of $\mathbf{J}$ have absolute value strictly less than 1, so long as both $\gamma$ and $N$ are not both equal, and the ratio remains strictly less than 1. □

## 6.2 Exact formula for $k = 2$

The analysis of the stability of the mean field equation for an arbitrary number of random variables runs into technical challenges, specifically the ordering of the expectations does not satisfy many of the usual inequalities of absolute differences. Some work in this are has been done. For this reason we study the Lyapunov stability of the mean field equation for bivariate normal random variables. We use a special form of the expected maximum for bivariate random variables that can be derived directly from Equation (11), but can also be found in (Clark, 1961), (Ker, 2001) and (Ross, 2003).

**Proposition 2.** The mean field equation (2) is stable around an equilibrium point when the action space of our MDP has cardinality 2; $|A| = 2$, and the discount factor $\gamma$ is strictly less than 1.

*Proof.* Suppose we have a random variable $\mathbf{X} = (X_1, X_2) \sim \mathcal{N}_2(\boldsymbol{\mu}, \boldsymbol{\Sigma})$ with $X_1 \perp X_2$, we can write

$$\mathbb{E}(\max_i X_i) = \mu_1 \Phi\left(\frac{\mu_1 - \mu_2}{\alpha}\right) + \mu_2 \Phi\left(\frac{\mu_2 - \mu_1}{\alpha}\right) + \alpha \phi_1\left(\frac{\mu_2 - \mu_1}{\alpha}\right) \tag{41}$$

where $\alpha = \sqrt{\sigma_1^2 + \sigma_2^2 - 2\Sigma_{1,2}}$. Putting this expression into the mean field equation and then taking a partial derivative with respect to the mean of the first variable we have

$$\frac{\gamma}{N} \frac{\partial}{\partial \mu_{s',a_1}^t} \left[ \mu_{s',a_1}^t \Phi\left(\frac{\mu_{s',a_1}^t - \mu_{s',a_2}^t}{\alpha}\right) + \mu_{s',a_2}^t \Phi\left(\frac{\mu_{s',a_2}^t - \mu_{s',a_1}^t}{\alpha}\right) + \alpha \phi_1\left(\frac{\mu_{s',a_2}^t - \mu_{s',a_1}^t}{\alpha}\right) \right]. \tag{42}$$

Consider the partial derivative of the first cumulative distribution function. We have

$$\frac{\partial}{\partial \mu_{s',a_1}^t} \Phi\left(\frac{\mu_{s',a_1}^t - \mu_{s',a_2}^t}{\alpha}\right) = \frac{\partial}{\partial \mu_{s',a_1}^t} \int_{-\infty}^{\frac{\mu_{s',a_1}^t - \mu_{s',a_2}^t}{\alpha}} \frac{1}{\sqrt{2\pi}} e^{-\frac{1}{2}t^2} \, dt \tag{43}$$

$$= \frac{1}{\alpha\sqrt{2\pi}} \exp\left\{ -\frac{1}{2}\left(\frac{\mu_{s',a_1}^t - \mu_{s',a_2}^t}{\alpha}\right)^2 \right\}. \tag{44}$$

From which it is easy to see that the derivative of the second cumulative distribution function is

$$\frac{\partial}{\partial \mu_{s',a_1}^t} \Phi\left(\frac{\mu_2 - \mu_1}{\alpha}\right) = -\frac{1}{\alpha\sqrt{2\pi}} \exp\left\{ -\frac{1}{2}\left(\frac{\mu_{s',a_2}^t - \mu_{s',a_1}^t}{\alpha}\right)^2 \right\} \tag{45}$$

and for the other derivative with respect to $\mu_{s',a_2}^t$ we have

$$\frac{\partial}{\partial \mu_{s',a_2}^t} \Phi\left(\frac{\mu_2 - \mu_1}{\alpha}\right) = \frac{1}{\alpha\sqrt{2\pi}} \exp\left\{ -\frac{1}{2}\left(\frac{\mu_{s',a_2}^t - \mu_{s',a_1}^t}{\alpha}\right)^2 \right\} \tag{46}$$

$$\frac{\partial}{\partial \mu_{s',a_2}^t} \Phi\left(\frac{\mu_1 - \mu_2}{\alpha}\right) = -\frac{1}{\alpha\sqrt{2\pi}} \exp\left\{ -\frac{1}{2}\left(\frac{\mu_{s',a_1}^t - \mu_{s',a_2}^t}{\alpha}\right)^2 \right\}. \tag{47}$$

The derivative of the probability density function is simply

$$\frac{\partial}{\partial \mu_{s',a_1}^t} \phi\left(\frac{\mu_{s',a_2}^t - \mu_{s',a_1}^t}{\alpha}\right) = \frac{1}{\sqrt{2\pi}} e^{-\frac{(\mu_{s',a_2}^t)^2}{2\alpha^2}} \cdot \frac{\partial}{\partial \mu_{s',a_1}^t} \left[ e^{-\frac{(\mu_{s',a_1}^t)^2}{2\alpha^2}} e^{\frac{\mu_{s',a_1}^t \mu_{s',a_2}^t}{\alpha^2}} \right] \tag{48}$$

where

$$\frac{\partial}{\partial \mu_{s',a_1}^t} \left[ e^{-\frac{(\mu_{s',a_1}^t)^2}{2\alpha^2}} e^{\frac{\mu_{s',a_1}^t \mu_{s',a_2}^t}{\alpha^2}} \right] = \frac{\mu_{s',a_2}^t}{\alpha^2} e^{-\frac{(\mu_{s',a_1}^t)^2}{2\alpha^2}} e^{\frac{\mu_{s',a_1}^t \mu_{s',a_2}^t}{\alpha^2}} - \frac{\mu_{s',a_1}^t}{\alpha^2} e^{-\frac{(\mu_{s',a_1}^t)^2}{2\alpha^2}} e^{\frac{\mu_{s',a_1}^t \mu_{s',a_2}^t}{\alpha^2}}. \tag{49}$$

We simplify notation, letting $\mu_{s',a_1}^t = \mu_1$ and $\mu_{s',a_2}^t = \mu_2$, and writing $g_{12} = \frac{\mu_1 - \mu_2}{\alpha}$ and $g_{21} = \frac{\mu_2 - \mu_1}{\alpha}$. This allows us to write the full partial derivative with respect to $\mu_1$ as:

$$\frac{\gamma}{N} \sum_{s'} \left[ \Phi(g_{12}) + \frac{1}{\alpha\sqrt{2\pi}} \left( \mu_1 e^{-\frac{1}{2}g_{12}^2} - \mu_2 e^{-\frac{1}{2}g_{21}^2} \right) + \frac{\alpha}{\sqrt{2\pi}} \left( \frac{\mu_2}{\alpha^2} - \frac{\mu_1}{\alpha^2} \right) e^{-\frac{1}{2}(\mu_1 - \mu_2)^2} \right] \tag{50}$$

The corresponding partial derivative with respect to $\mu_2$ is

$$\frac{\gamma}{N} \sum_{s'} \left[ \Phi(g_{21}) + \frac{1}{\alpha\sqrt{2\pi}} \left( \mu_2 e^{-\frac{1}{2}g_{21}^2} - \mu_1 e^{-\frac{1}{2}g_{12}^2} \right) + \frac{\alpha}{\sqrt{2\pi}} \left( \frac{\mu_1}{\alpha^2} - \frac{\mu_2}{\alpha^2} \right) e^{-\frac{1}{2}(\mu_1 - \mu_2)^2} \right]. \tag{51}$$

Writing Equation (50) at the fixed point with $A$, and Equation (51) at the fixed point with $B$, the Jacobian matrix and characteristic equation are:

$$\begin{bmatrix} A & B \\ A & B \end{bmatrix}; \quad \lambda^2 - \lambda(A + B) = 0 \tag{52}$$

with eigenvalues of 0 and $A + B$. In the notation defined above, $A + B = \gamma/N \cdot \sum_{s'} [\Phi(g_{12}) + \Phi(g_{21})]$, which is further expanded as

$$A + B = \frac{\gamma}{N} \sum_{s'} \left[ \Phi\left( \frac{\mu_{s',a_1}^* - \mu_{s',a_2}^*}{\alpha} \right) + \Phi\left( \frac{\mu_{s',a_2}^* - \mu_{s',a_1}^*}{\alpha} \right) \right]. \tag{53}$$

It is clear that the expression inside the summation can be written as

$$\Phi(g_{12}) + \Phi(-g_{12}) = 1 - \frac{1}{\sqrt{2\pi}} \int_{-g_{12}}^{g_{12}} e^{-\frac{1}{2}t^2} \, dt \tag{54}$$

assuming $g_{12} \geq 0$. If $g_{21} \leq 0$ we integrate over $[-g_{21}, g_{21}]$ instead. It is clear that (54) is bounded above by 1 (true when $g_{21} \leq 0$ also), which implies

$$A + B = \frac{\gamma}{N} \sum_{s'} \left[ \Phi\left( \frac{\mu_{s',a_1}^* - \mu_{s',a_2}^*}{\alpha} \right) + \Phi\left( \frac{\mu_{s',a_2}^* - \mu_{s',a_1}^*}{\alpha} \right) \right] \tag{55}$$

$$\leq \frac{\gamma}{N} \sum_{s'} 1 \tag{56}$$

$$= \gamma. \tag{57}$$

Therefore, so long as the discount factor is strictly less than 1, the 2-action space mean field equation is stable. $\square$

## 6.3 Gumbel distribution

Consider a set $X_1, ..., X_k$ of Normal IID random variables. As the number of random variables increases to infinity, that is, as $k \to \infty$, the cumulative distribution function of the maximum of this set of random variables tends to the cumulative distribution function of the Gumbel distribution, see (Von Mises, 1936) and (Fisher and Tippett, 1928).

Define constants $a_n = 1/n\phi(b_n)$ and $b_n = \Phi^{-1}(1 - 1/n)$, then

$$\lim_{k \to \infty} F(a_n x + b_n)^k = e^{-e^{-x}} = G(x; 0, 1) \tag{58}$$

where $G(x; 0, 1)$ is the cumulative distribution function of the Gumbel distribution with location parameter $\nu = 0$ and scale parameter $\beta = 1$. Therefore if we have IID Normal random variables, constants exist such that

$$Pr[\max_i X_i \leq x] = \prod_i Pr[X_i \leq x] = F(x)^k \to_{k \to \infty} G(x; \nu, \beta). \tag{59}$$

We can absorb the constants $a_n$ and $b_n$ into the random variables $X_i$ and so maintain convergence to the standardized Gumbel distribution.

**Proposition 3.** As the number of actions tends to infinity, the mean field equation (2) is stable.

*Proof.* As the number of actions in our MDP model tends to infinity, $\max_a Q^t(s,a)$ becomes a Gumbel distributed random variable, and hence its expected value is just the mean value $\nu + \beta\gamma_{\text{em}}$, where $\gamma_{\text{em}}$ is our notation for the Euler-Mascheroni constant. With $\nu = 0$ and $\beta = 1$ we can substitute this into the mean field equation as

$$\mu_{s,a}^{t+1} = \mu_{\rho_{s,a}} + \gamma \sum_{s'} \bar{P}_{s,a,s'} \cdot \gamma_{\text{em}} \tag{60}$$

$$= \mu_{\rho_{s,a}} + \frac{\gamma}{N} \cdot N\gamma_{\text{em}} \tag{61}$$

$$= \mu_{\rho_{s,a}} + \gamma \cdot \gamma_{\text{em}}. \tag{62}$$

It is easy to see that all partial derivatives will be zero, hence all eigenvalues of the Jacobian matrix will be zero, that is, strictly less than 1 and hence stable. □

## 7 Contraction mappings

Throughout this section we use the standard definition of a contraction mapping defined on a metric space: Given a map $F : X \to X$ on a metric space $X$, we call $F$ a contraction if for all $x, y \in X$, there exists some real $k \in [0, 1)$ such that

$$d_X(F(x), F(y)) \le k\, d_X(x, y) \tag{63}$$

where $d_X$ is the metric defined on $X$.

In this section we show that different forms of the mean field equation are contractions. We do this by considering the mean field equation over all state-action pairs as an operator

$$\mathbf{F}_{\text{DMFP}} : \mathbb{R}^{|S| \times |A|} \to \mathbb{R}^{|S| \times |A|}, \quad \boldsymbol{\mu}^{t+1} = \mathbf{F}_{\text{DMFP}}(\boldsymbol{\mu}^t).$$

Similar to the consideration of stability, we begin by using the upper bound (19) in place of the expected maximum. After this we consider the case when the number of action goes to infinity $k \to \infty$, and we have a Gumbel distributed random variable for our expected maximum.

**Proposition 4.** If the expected maximum term in the mean field equation is substituted with the upper bound (19), then this form of the mean field equation is a contraction.

*Proof.* Let us write $\nu^t(s) = \max_a \mathbb{E}(Q^t(s,a)) = \max_a \mu_{s,a}^t$, and $(\nu')^t(s) = \max_a (\mu'_{s,a})^t$. The proof of the contraction property follows identically the one given by (Bertsekas, 2012), and found in (Agarwal et al., 2019). As in those proofs, we have

$$|\max_a \mu_{s,a}^t - \max_a (\mu'_{s,a})^t| \le \max_a |\mu_{s,a}^t - (\mu'_{s,a})^t|. \tag{64}$$

For a moment let us consider $\mu_{s,a}^{t+1}$ and $(\mu'_{s,a})^{t+1}$ from $\boldsymbol{\mu}^{t+1}$ and $(\boldsymbol{\mu}')^{t+1}$ respectively.

$$\mu_{\rho_{s,a}} + \gamma \sum_{s'} \overline{P}_{s'|s,a} \cdot \left( \max_{a'} \mu_{s',a'}^t + \sqrt{\frac{k-1}{k} \sum_{a'} (\sigma_{s',a'}^2)^t} \right) \tag{65}$$

$$\mu_{\rho_{s,a}} + \gamma \sum_{s'} \overline{P}_{s'|s,a} \cdot \left( \max_{a'} (\mu'_{s',a'})^t + \sqrt{\frac{k-1}{k} \sum_{a'} (\sigma_{s',a'}^2)^t} \right). \tag{66}$$

As we can see, the mean of the mean rewards is the same for both expressions. Additionally, the sum over the square root term is the same in both expressions, and since the summation is distributive, the sum over

the square root term will be the same in both expressions, hence when we take the absolute difference of the two we get

$$\left| \gamma \sum_{s'} \bar{P}_{s'|s,a} \cdot \max_{a'} \mu_{s',a'}^t - \gamma \sum_{s'} \bar{P}_{s'|s,a} \cdot \max_{a'} (\mu_{s',a'}')^t \right| \tag{67}$$

which we write succinctly as $\left| \gamma \bar{P}_{s,a} \nu^t(s) - \gamma \bar{P}_{s,a} (\nu')^t(s) \right|$. With this notation specified we can now prove the main result. Using the $\ell^\infty$-norm we have

$$\left\| \boldsymbol{\mu}^{t+1} - (\boldsymbol{\mu}')^{t+1} \right\|_\infty = \gamma \left\| \bar{P} \nu^t - \bar{P}(\nu')^t \right\| \tag{68}$$

$$\leq \gamma \left\| \nu^t - (\nu')^t \right\|_\infty \tag{69}$$

$$= \gamma \max_s | \nu^t(s) - (\nu')^t(s) | \tag{70}$$

$$\leq \gamma \max_s \max_a | \mu_{s,a}^t - (\mu_{s,a}')^t | \tag{71}$$

$$= \gamma \left\| \boldsymbol{\mu}^t - (\boldsymbol{\mu}')^t \right\|_\infty \tag{72}$$

which concludes the proof. $\qquad\square$

A trivial lower bound on the expected maxima that we have not covered so far is one we call the Jensen lower bound, since via the convexity of the maximum function, the expected maximum is greater than the maximum of the expected values, see (Feller, 1971). It is trivial to see that the mean field equation is a contraction when the expected maximum is substituted with the maximum of the expected values. As both the 'Aven' and Jensen mean field equation variants are contractions, they have fixed points, and since these variants bound the exact mean field equation, we know that the sequence of points of the exact mean field equation over iterations is bounded. If it could be shown that the sequence $\{\mu_{s,a}^0, \mu_{s,a}^1, ...\}$ is monotonic, this would also furnish a proof that the mean field equation, (2), has a fixed point, without resorting to a proof that it is a contraction.

Similar to the Jensen bound form of the mean field equation, the mean field equation where $k \to \infty$ and we use the Gumbel mean for the expected maximum is trivially a contraction. Looking at the Gumbel mean field equation from the previous section:

$$\mu_{s,a}^{t+1} = \mu_{\rho_{s,a}} + \gamma \cdot \gamma_{\text{em}}, \tag{73}$$

we can see that there is no dependency on the previous mean values, and so the difference between any two $\mu$'s will be zero. This satisfies the definition of a contraction for any value of $k \in [0,1)$.

Similar to our examination of stability, our results for contraction mappings are limited to a handful of special cases and estimations of the mean field equation. While extending these results to the general mean field equation is beyond the scope of this paper, we note that the cumulative evidence for this conjecture being true prompts us to study this in future work.

## 8 Final remarks

Improving methods of sample efficiency in Bayesian reinforcement learning is a difficult task. The mean field approach taken here approaches this problem from a different perspective, aiming to compute approximate statistics based on a theory that is asymptotically exact under certain mean field conditions. A general hope is that such methods may provide less biased and tighter estimates of the statistics over state-action value functions, which in turn should help in the design of new learning algorithms.

Our study has reviewed and clarified a suite of methods for computation or approximation of the expected maximum. We have measured the accuracy of different estimation formulas for the expected maximum of a set of Normal random variables, a comparative study that we have then shown significantly affects the computation of posterior statistics of the Bellman optimality equation. We have demonstrated the accuracy of the equations in the non-identical belief setting, under the mean field conditions and outside of these

conditions. Specifically, we have tested the performance of the DMFP equations as the Dirichlet beliefs reflect a higher degree of sparsity in the MDP transitions.

An open theoretical challenge, discussed in the preceding section, is to establish whether the DMFP equations for the discounted infinite horizon case, are in general contraction mappings just as the Bellman optimality equation is. This has proven a very challenging theorem to study and we look forward to any attempts to establish the result or to be provided counterexamples. Alternatively, one may attempt to establish the dynamical stability of DMFP in general.

Other possible challenges for this line of work include investigating the performance of the mean field equation approximations we have developed here during the course of learning. For the highly connected MDPs discussed in 4, the mean field theory will hold throughout learning, but as sparsity is introduced this will no longer hold, as shown. More advanced statistical field theories, beyond mean field theory, will likely require the use of the analytic and approximate forms for the expected maximum used here. Specifically, we anticipate that second and higher order corrections to the "first order" DMFP will require derivatives of the expected maximum (Helias and Dahmen, 2020). It is these corrections which may assist in the construction of new upper confidence bound type Bayesian algorithms.

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
