# OpenReview forum: "Expectation of the maximum of random variables with applications to reinforcement learning"
_TMLR — Rejected by TMLR_

### Review · Reviewer_t1iu · 2023-07-24

**Summary Of Contributions:**

This paper proposes the use of dynamic mean field programming (DMFP) for Bayesian reinforcement learning, in particular for multi-armed bandit RL. The authors present a broad overview of some of the DMFP theory that is relevant, provide numerical simulations of the accuracy of various approaches, and present some stability and contraction analyses on these methods.

**Audience:**

Yes

**Claims And Evidence:**

Yes

**Requested Changes:**

# Suggestions / comments/ questions
1. Please use `\citep` and `\citet` explicitly. Right now it seems it's using `\citet` by default, which renders some things incorrectly (e.g. "such as finite size corrections Helias and Dahmen (2020)." should really be "such as finite size corrections (Helias and Dahmen; 2020)."
1. Please use hyperref so that citations, equations, and sections are clickable. This makes it easier to navigate when reading digitally.
1. In section 2.1 you define $R$ as $R:S\times A\rightarrow\mathbb{R}$, which is _not_ a distribution, yet you discuss rewards as if they _are_ a distribution.
1. In the third paragraph of section 2.2, you say "The core idea that the mean field theory brings to helping solve this problem is that, in the large state space limit, the correlations between the Q-value functions are effectively zero.", but it's not clear why this is the case. Please elaborate.
1. In **Discussion of DMFP theory.**, is the fact that the mean value can be written as $\mu_{\rho_{s,a}} + C$ a consequence of the independence assumptions? Please be more clear and explicit about this.
1. In section 3 it would be useful to add pointers to the specific sections where each of the points are discussed.
1. RIght before equation (6), specify that $X_1,\ldots, X_k$ have CDFs $F_{X_1},\ldots,F_{X_k}$, as it's necessary in equation (6).
1. What are the $X$s (with no subscripts) in equation (14)?
1. What is $erf$ in equation (15)?
1. In section 3.4 you're referencing numerical experiments run. Are these in the paper? If so, add links to them. If not, why not?
1. All the numerical results (tables and figures) are missing confidence intervals (CIs). Please add them, as these are critical for assessing whether the differences are significant or not.
1. Do you really need 10 digits of precision for the numerical results? I think CIs would be more effective.
1. What do the bolded numbers mean in the tables?
1. In Algorithm 1 neither $\bar{P}$ nor $\mu_{\rho}$ are updated in the algorithm.
1. In Algorithm 1, why do you need mean estimates for each iteration? In other words, couldn't we just have the mean table be $(N\times K)$?
1. In Algorithm 1, in the first nested for loop, should the $\mu$ be $\mu_{\rho}$?
1. In Algorithm 1, in the second for loop, where is $Q$ coming from?
1. In the second half of the second for loop, you use $P$, but should this be $\bar{P}$?
1. In the paragraph below Table 5, the first sentence seems to contradict the following sentence (e.g. (11) seems more accurate than (6)).
1. In section 4.2, why do you only evaluate equation (19) in addition to (6)?
1. In Figure 6, what are the axes?
1. In section 5 you say "the role that the DMFP can provide in other reinforcement learning algorithms, besides Q-learning.", but you haven't really shown the case for Q-learning.
1. In section 5, you mention BOLT but haven't introduced it yet.
1. In equation (25), what is $\beta$?
1. In equation (25), the transition function has an extra parameter $\phi$, but it hasn't been explained how this affects the transitions. Same point for equation (29).
1. At the bottom of page 15 you say "especially in the large system limit, where the mean field equation approaches perfect numerical accuracy.", but both methods evaluated in Figure 8 seem to converge at the same point (iteration 20)?
1. What is the y-axis in Figure 8?
1. There are lots of terms that have not been introduced in equation (34), so it's hard to follow.
1. In Proposition 1 you say "iterative function is stable around an equilibrium point is stable provided the ratio $\gamma / N$ is strictly less than 1.", but isn't this always the case?



# Minor suggestions
1. There is an extra "the" in the abstract
1. Right below equation (17) it should be "**In** general in discrete..."
1. In Figure 6, I think you mean (19) in the legend
1. In the second line of section 6.3 I think it should be $k\rightarrow\infty$

**Strengths And Weaknesses:**

# Strengths

The paper is clearly written, and the overview of DMFP is quite good and will be useful for those unfamiliar with that field.
The theoretical results are clearly presented and the numerical experiments are effective.

# Weaknesses

The biggest weakness for me is that I don't feel it's sufficiently well contextualized. There is some discussion in the introduction on how all the results presented relate to Bayesian RL, but for the most part it reads as a long stream of results that seem to exist in isolation. Perhaps adding a section that fleshes out the aplication of DMFP to RL would help. Further, the "Final remarks" section could be expanded quite a bit more. This would help clarify and contextualize the work, suggest avenues for future research, and discuss limitations of the current work and open problems.
One high-level question to consider addressing in the paper is: Who should consider applying these methods, and in what problems/situations? As I said above, this is discussed a little in the introduction, but it feels too brief to me.

If you expand and make more precise the second paragraph of section 2.2, this will also aid in this contextualization issue.

In each section it would also be useful to add contextualizing guidance for the reader. For instance, in section 3, before diving into the technical results, remind the reader _why_ we care about expectations, _why_ indpedence and normal variables are a reasonable assumption, etc.

---

> ### Author Response · Authors · 2023-09-22
> **Response to Reviewer t1iu**
>
> We thank the reviewer for their comments, we appreciate the constructive criticism, and believe we have improved the paper as suggested in the revision.
>
> We begin by addressing the "weaknesses" section of the review. The responses to the detailed requested changes are below.
>
> ## Weaknesses
>
> ### Context for mean field theory for RL, "adding a section that fleshes out the application of DMFP to RL would help"
>
> We agree that the improve would benefit from adding a section fleshing out the application of DMFP to RL, as well as expanding the second paragraph in section 2.2 and section 3, and the Final remarks. The revision we have uploaded contains these changes. In particular we point the reviewer to the revised introduction and section 2. We also point to the new simulations and improved discussion in section 4 and 5 that follow from this improved focus.
>
> As a general response, our changes emphasise that the motivation for our work is to better estimate the statistics of the optimal Q-values. To quote O'Donoghue et al. 2016, from their paper "The Uncertainty Bellman Equation and Exploration":
>
> >*``If we had access to the true posterior distribution over the Q-values then we could take actions that lead to states with higher uncertainty by, for example, using Thompson sampling (Thompson, 1933; Strens, 2000), or constructing Q-values that are high probability upper bounds on the true Q-values and using the OFU principle (Kaufmann et al., 2012). However, calculating the true posterior is intractable for all but very small problems. Due to this difficulty prior work has sought to approximate the posterior distribution (Osband et al., 2017), and use that to drive exploration."*
>
>
> ### "Who should consider applying these methods, and in what problems/situations?"
>
> As the quote from O'Donoghue suggests, if we could estimate the moments of Q-values, we could design better exploration bonuses! As such, the approach is generic, though it may not apply to all MDPs. Future work involves quantifying exactly which MDPs it works for, and how the error breaks down outside of the 'ideal' mean field conditions. Our new experiments in Section 4 begin to explore this question.
>
> We argue that although the mean field theory is only exactly true under certain conditions, it can be thought of as a very good approximation outside these conditions. This is just like variational approximation idea in Bayesian inference (eg minimising KL divergence between true and approximate posterior).
>
> It may assist some readers to mention that the mean field equations we study are a new type of message passing for approximate Bayesian inference. A prominent example of message passing includes the Belief Propagation algorithm for probabilistic graphical models. We describe this analogy in the paper, and we are unaware of other methods that are derived in a principled manner (eg. from an appropriate energy function).
>
> We clarify the structural assumptions of general MDPs and how this relates to the conditions mean field theory. The first work on DMFP theory focused exclusively on the flat Dirichlet and identical mean-reward case. The present paper generalises to the case where mean-rewards are arbitrary. Unlike the DMFP work, we move from a single mean field equation to a full set of $N\times A$ mean field equations.
>
> Our new experiments reveal what the effect of sparsity is on the mean field approximation quality. The promising results is that the DMFP approximation breaks down gracefully as the beliefs over MDP reflect more sparsity in the MDP transition probabilities.
>
>
> Ref: O’Donoghue, Brendan, et al. "The uncertainty bellman equation and exploration.", ICML 2018.

---

> > ### Author Response · Authors · 2023-09-22
> > **Reply (continued)**
> >
> > ## Requested Changes:
> >
> > 1. Completed
> >
> > 2. Completed
> >
> > 3. Thank you, we have clarified that the rewards are random variables.
> >
> > 4. Thank you, we have attempted to clarify this in the new revision. The contribution of any single variable to the value of any other becomes negligible as the state space becomes sufficiently large. In the Bellman equation, at each iteration or time step, each Q-value can directly depend ("couple") to all other Q-values, via the sum over possible next states. The genesis of a mean field is to then ask the question: How much of the value $Q^{t+1}_{(s,a)}$ is determined by the value of one of the $Q^t{s',a'}$ values in the sum term? The answer is that  *if we believe the transitions connect to many other states* the contribution of any single such value is very small, and gets smaller as the state space grows. The paper of [Stamatescu2022] makes this precise via a saddlepoint approximation.
> >
> > 5. Yes under the mean field setting that is true. However, outside of the mean field settings, the DMFP equations will not be of the form  $\mu_{\rho_{s,a}} + C$, and the constant $C$ will depend on the state-action pair, in general.
> >
> > 6. Completed
> > 7. Completed
> > 8. They are standard Normal pdf and cdf, respectively. Paper has been edited to include this information.
> > 9. erf, is the error function. This has been noted in the paper and a reference added.
> > 10. Edited paper to include a sentence referencing simulations in section 4.
> > 11 - 13. All of these questions refer to the simulations and whether or not we should add confidence intervals, if the numerical precision (10 decimals) is necessary, and the meaning of the bold numbers. We have updated the numerical simulations signficantly in light of your queries.
> > 14. These do not update, the DMFP calculate means of the Q-values given estimates of the MDP parameter means.
> > 15. It is precisely the means of the Q-values which we are estimating in this routine.
> > 16. Fixed.
> > 17. This algorithm pseudo-code has been clarified, to correspond to the DMFP equations.
> > 18. Fixed.
> > 19. This error has been corrected, the Afonja and Ross approaches agree.
> > 20. The reason that we only considered the Ross equation and the Afonja equation is because out of the five DMFP alternative we evaluated for accuracy previously, Afonja cannot generalise to $k = 10$ actions without significantly convoluted programming, and Bert MAX and Bert MIN both come out to being very close to the Aven DMFP equation.
> > 21. The axes in Figure 6., are for the density and value of the Q-values, respectively. This is now Figure 4.
> > 22. Thank you, we meant to refer to $Q$-value iteration not $Q$-Learning.
> > 23. We have removed mention of BOLT.
> > 24. The $\beta$ in equation (25) is a parameter, upon which the main theorem of the BEB depends to attain the  PAC-Bound. This depends on the confidence level desired for the bound.
> > 25. We have clarified this and made it consistent with the notation earlier in the paper.
> > 26. These agree because the RL algorithms considered here are in the finite horizon case, and at the end point (with one step remaining) the Q-values are precisely the mean-rewards, and hence there is no error in BEB, since from this point it is not an MDP.
> > 27. The y-axis in Figure 8 is the initial value for the value function
> > 28. We have introduced properly all the terms in the K-Learning equation.
> > 29. Yes, in general the discount factor can be any real value in $[0,1)$, and this is what we meant implicitly, we have corrected this.

---

> > > ### Comment · Reviewer_t1iu · 2023-09-27
> > >
> > > Thank you for your response and your changes.
> > >
> > > The introduction and sections 2 and 3 are reading _much_ better now, and they do a good job at contextualizing the work.
> > >
> > > One thing I'd recommend for the future is to use a different colored text for the changes you made, as it makes it easier for reviewers to know where to focus on.
> > >
> > > For this paper, I would also recommend to do another editing pass on the new text written to get rid of any grammatical or spelling mistakes. For instance:
> > > - in the second paragraph of section 1.1 there is: "In practice however, such translations have result**ed** in both..."
> > > - in the second paragraph of section 3 there is "especially in the discipline of **cmputert** science"
> > > - etc.

---

### Review · Reviewer_PrJY · 2023-08-08

**Summary Of Contributions:**

This work considers the task of computing an approximately optimal policy in the Bayesian RL setting, i.e., when there is a prior distribution the MDP is assumed drawn from (unlike the minimax setting). It posits that under certain condition the state-action values (random themselves because the MDP is random) are independent across states. The work then tours a sequence of upper bounds on the supremum of a stochastic processes (often Gaussian), and studies the result of using them in Bellman updates.

**Audience:**

No

**Broader Impact Concerns:**

NA.

**Claims And Evidence:**

No

**Requested Changes:**

Please see above.

**Strengths And Weaknesses:**

Computation of the exact Bayesian optimal policy is often, outside of a few exceptions, e.g., Gittins, intractable. While relaxed formulations (via Bernstein, e.g.) do lead to vanishing average regret, there remains the possibility of obtaining better approximations using tighter characterizations of posterior updates than those made possible by tracking the first few moments. In this aspect, I find the underlying motivation of the work to be one that’s well aligned with the Bayesian RL literature.

However, throughout the paper, the exposition makes a number of logical leaps without adequate justification. These leaps, often in the form of assumptions or particulars of the problem setting, cast doubt on how generally applicable the approach is, and if it ultimately delivers better policies in the Bayesian RL setting.

To start with, the writing shies away from making (rigorously) precise when the assumption here apply to an MDP at hand. What structural properties must an MDP satisfy to lend itself the stated analysis? There’s a reference to a “large, highly connected MDP”; do we measure this connectedness by some measure of conductance, shortest (expected) path, or some metric altogether different? What happens if the transition functions for some fraction of states are near-deterministic (delta-function-like)? The paper attempts to shift the burden of such justification entirely to prior work. This poses a couple pf problems— given the present exposition, it is difficult for the reader to ascertain if the assumptions are reasonable; more crucially, while a (deferred justifications) strategy is perhaps defensible if the prior work is well accepted by the community at large either via qualitative or quantitative metrics (e.g., citations, say), I find it hard to make such a case here.

The Bellman equation in the Bayesian setting (e.g., see Definition 4.1 in https://arxiv.org/pdf/1609.04436.pdf) while maximizing (point wise) for action must also account change in beliefs induced by the state transition, in addition to the value at the successor state. In fact, this is the sole driver of exploration in the Bayesian setting. Is there a reason for absence of such considerations here (e.g., in equations 1 and 2)?

The present development spends a consider amount of time on suprema of Gaussian processes. Is the reason of this analytic tractability? Or, perhaps a more use-oriented (or CLT-esque) justification?

It’s hard to assess the fidelity of synthetic examples here to common considerations/benchmarks in RL.

---

> ### Author Response · Authors · 2023-09-21
> **Response to Reviewer PrJY**
>
> We thank the reviewer for their comments, we appreciate the candour and constructive criticism. We address all points carefully below, and all clarifications have been incorporated into the submitted revision. We now address the reviewers specific comments and questions.
>
> 1. "Computation of the exact Bayesian optimal policy is often, outside of a few exceptions, e.g., Gittins, intractable... In this aspect, I find the underlying motivation of the work to be one that’s well aligned with the Bayesian RL literature."
> " The Bellman equation in the Bayesian setting (e.g., see Definition 4.1 in https://arxiv.org/pdf/1609.04436.pdf) while maximizing (point wise) for action must also account change in beliefs induced by the state transition, in addition to the value at the successor state. In fact, this is the sole driver of exploration in the Bayesian setting. Is there a reason for absence of such considerations here (e.g., in equations 1 and 2)?}"
>
> Here we do not consider the RL problem via the "Bayes adaptive" approach. Our work considers only current beliefs (based on statistics gathered), based on observed rewards and transitions, not in anticipation of new observations. Our work  is relevant instead to the asymptotically optimal literature (eg. UCB algorithms that attain sublinear expected cumulative regret) which are only heuristically anticipative by adopting the "optimism" principle (see Chang and Lai, and more recently Russo for the equivalence between the two for infinitely patient Gittins agents).
> One can take the Bayesian or frequentist approaches to estimation, whether one is anticipative of future information is an orthogonal choice. Hence, our equations 1 and 2 are to be based entirely on past data, and thus only the current belief state.
>
> 2. " the exposition makes a number of logical leaps without adequate justification. These leaps, often in the form of assumptions or particulars of the problem setting, cast doubt on how generally applicable the approach is, and if it ultimately delivers better policies in the Bayesian RL setting. "
>
> The application of our study and future work is well described by a quote from  O'Donoghue et al, "The Uncertainty Bellman Equation and Exploration":
>
> "If we had access to the true posterior distribution over the Q-values then we could take actions that lead to states with higher uncertainty by, for example, using Thompson sampling (Thompson, 1933; Strens, 2000), or constructing Q-values that are high probability upper bounds on the true Q-values and using the OFU principle (Kaufmann et al., 2012). However, calculating the true posterior is intractable for all but very small problems. Due to this difficulty prior work has sought to approximate the posterior distribution (Osband et al., 2017), and use that to drive exploration."
>
> This intractability for the Bellman equation is due to its general dependence structure between value functions. Our paper addresses open questions in the relatively new area of using mean-field theories for RL, where we aim to approximate directly the mean and variance (and higher moments) of the optimal Q-values.
>
> Note that the method of O'Donoghue et al. is only for policy evaluation, a much simpler equation and one with less utility than the optimality equation. Moreover, it resorts to very loose bounds.
>
> 3. "What structural properties must an MDP satisfy to lend itself the stated analysis? There’s a reference to a “large, highly connected MDP”; do we measure this connectedness by some measure of conductance, shortest (expected) path, or some metric altogether different? What happens if the transition functions for some fraction of states are near-deterministic (delta-function-like)?"
>
> The DMFP theory was derived under the assumptions we detail in the revised section 2.2. In the new section 2.3 we detail the new settings we consider in this paper. Regarding your question on connectedness versus "delta-like transitions", the revised paper clarifies this point, and presents new simulations which consider very sparse delta-like transitions.
>
> The DMFP theory holds for  $\alpha_{s'|sa} = c$, with c greater or equal to 1, for all (s',s,a). If the transitions are very sparse, and the $\alpha$ are also sparse (eg. many 1's and then some alpha with very large values), then we would expect the mean field theory not to hold. We have simulated this setting exactly, and we see that the mean field theory breaks down quite gracefully, as measured by the error in the new DMFP equations derived here.

---

> > ### Author Response · Authors · 2023-09-21
> > **Further reponse**
> >
> > 4. "The present development spends a consider amount of time on suprema of Gaussian processes. Is the reason of this analytic tractability? Or, perhaps a more use-oriented (or CLT-esque) justification?"
> >
> > The reason we emphasised Gaussian random variables was because of the availability of exact results. However, as our results show, the best approximations hold for arbitrarily distributed independent random variables, ie. the Ross approximation based on numerical integration.
> >
> > Note that however, there is in a sense a "CLT-esque" justification for the mean field theory, and higher moments of the mean field theory (higher order expansions of the generating function of [Stamatescu 2022]) will approximate the posterior correlations over Q-values with a Gaussian. This will result in mixture distributions for the Q-values, see [Stamatescu2022] and M. Helias and D. Dahmen "Statistical Field Theory for Neural Networks, 2016,  for further details.
> >
> > 5. "It’s hard to assess the fidelity of synthetic examples here to common considerations/benchmarks in RL."
> >
> > In the revised paper, we have included new simulations in Section 4 and 5. However, we are investigating a new line of research stemming from the discovery that the Q-values become independent under the newly developed mean-field theory. A new set of benchmarks could be devised accordingly, with this setting perhaps amongst the "easiest" class of MDPs.
> >
> > We thank the reviewer again for their time.

---

### Review · Reviewer_pxis · 2023-09-11

**Summary Of Contributions:**

This paper studies the computation of the expected Q-values of large state space MDPs under the Bayesian framework, using a recently proposed dynamic mean-field programming (DMFP) approach. The authors propose various methods for estimating the expectation of max over (different action components of) Q-values that appear in the DMFP computation. Some numerical experiments are conducted to compare the various proposed estimation methods as well as some baselines such as naive Monte Carlo and some Q-value updates adapted from existing Bayesian RL algorithms. Lyapunov stability and contractivity of the DMFP mappings are also derived for certain special cases of expected maxima estimations.

**Audience:**

Yes

**Claims And Evidence:**

No

**Requested Changes:**

Here are some major changes, comments and questions that need to be addressed.
* First and foremost, Bayesian RL is concerned with getting some policies that perform well, evaluated in terms expected total trajectory rewards, or its regret and sample complexity. It’s unclear why would we care about the expected Bayesian Q-values. In fact, many Bayesian RL algorithms such as PSRL directly samples an MDP from the posterior distribution and then solves it to get the policies to be executed and evaluated. Also, the two algorithms BEB and VBRB compared in this paper also directly address the sample complexity of the resulting policies in terms of the total trajectory rewards. The authors should explain clearly how estimating the exact expected Bayesian Q-values can be helpful for getting good policies in Bayesian RL, as if we look at the current paper without the Bayesian RL components (including DMFP, etc.), then the expected maxima computations are just summary of existing classical probability results.
* The assumptions in this paper are not very clear. I would suggest the authors to focus on a single setting (e.g., when the mean reward is independent and normal) throughout the paper, and leave more general results as a later section or appendix. Also, in the DMFP algorithm/update, we need the distribution properties of Q_t to proceed with the next iteration t+1, but this is not clearly explained. For example, if we want to use the formula in Section 3 that require normal and/or independent distributions, we need to show that Q_t is normal and/or independent by induction, which is unclear whether to hold or not from the current paper. On a related point, in the contractivity analysis, it’s not clearly stated or explained whether the fixed point set of the original exact mean-field equation (2) is the same as those approximate ones based on upper/lower bounds, etc., and if not, what the errors are. Without such statements, the contractivity can be meaningless.
* The comparison with BEB and VBRB in the numerical experiments is unfair. In fact, as mentioned above, the goal of the BEB paper is to use (25) to find the policy to be executed and evaluated, instead of estimating Q functions. This is also similar for VBRB in Sorg et al. (2012).

There are also some more minor issues, as listed below (roughly in the order they appear in the paper).
* In the abstract, there is an extra “the” in line 8. It’s not very clear what “current approaches” mean on line 10. Also, it’s unclear what “at the start of learning” means.
* I don’t see how “prior” and “posterior” make a difference in the DMFP updates, as throughout the paper the distributions of P and r serve as a background and the computation and updates of P and r are not discussed.
* It would be better to clearly state the large state space limit assumption in a rigorous manner when introducing (2). Also some pointers of the cited results (e.g., the example at the bottom of page 4) to the corresponding results/pages in the original DMFP paper would also be helpful.
* At the bottom of page 4, the comment that “we should expect to see that the mean value will be equal to $\mu_{\rho_{s,a}}$ + C for some constant amount C” makes it more questionable how estimating the expected Bayesian Q-values can be useful in Bayesian RL. In fact, this seems to suggest that if we want to execute the policies from the posterior mean of the Bayesian Q-values, it only depends on the instantaneous/one-step reward means.
* On top of page 5, what does “the same way” mean?
* At the beginning of Section 5, it seems that “… comparing to a finite …” should be “… comparing to is for a finite …”. Also, it’s not explained what $\phi$ is and why it equals $1/N$ in this paper.
* In Section 5, maybe I missed something but why Afonja, etc. are not compared (but only see “Ross’)?

**Strengths And Weaknesses:**

Strengths:
* Studies different expected maxima estimation methods under different assumptions and their application to the DMFP updates.
* Derived the stability and contractivity results for the resulting DMFP updates under some special cases of expected maxima estimations.
* Conducted numerical experiments to validate the theoretical findings and the performance of different estimation methods.

Weaknesses:
* The meaningfulness of computing the expected Q-values of MDPs under the Bayesian framework is questionable from the starting point. At least some more references need to be provided to support this.
* The writing, clarity and rigorousness need substantial improvement.
* The numerical experiments that compare with existing RL algorithms is not convincing, and actually those RL algorithms are not event proposed for estimating the expected Bayesian Q-values at all.

---

> ### Author Response · Authors · 2023-09-22
> **Response to Reviewer pxis**
>
> We thank the reviewer for their very clear and focussed review. We have substantially revised the manuscript and believe we have addressed the weaknesses identified, and made the requested changes. Our response will first address the weaknesses, then go through the requested changes by item.
>
> ## Weaknesses:
>
> ### "The meaningfulness of computing the expected Q-values of MDPs under the Bayesian framework is questionable from the starting point. At least some more references need to be provided to support this."
>
> We have positioned our paper more clearly, with more references justifying the study. Our motivation is well described by a quote from  O'Donoghue et al. 2018 "The uncertainty Bellman Equation",
>
> > *"If we had access to the true posterior distribution over the Q-values then we could take actions that lead to states with higher uncertainty by, for example, using Thompson sampling (Thompson, 1933; Strens, 2000), or constructing Q-values that are high probability upper bounds on the true Q-values and using the OFU principle (Kaufmann et al., 2012). However, calculating the true posterior is intractable for all but very small problems. Due to this difficulty prior work has sought to approximate the posterior distribution (Osband et al., 2017), and use that to drive exploration."*
>
> This intractability for the Bellman equation is due to its general dependence structure between value functions. Under the mean field setting this dependence simplifies. Our paper aims to first calculate the mean of the optimal Q-value function, for more general (non-identical) beliefs over MDP parameters. The higher central moments (variance and higher) at the first-order mean field theory are just those of the mean-reward. Future work will correct the mean field theory, to produce new estimates for variance (and covariance) of the optimal Q-values.
>
> ### "The writing, clarity and rigorousness need substantial improvement."
>
> We have revised the writing and clarity, and have also stated the assumptions of the mean field theory precisely.
>
> ### "The numerical experiments that compare with existing RL algorithms is not convincing, and actually those RL algorithms are not event proposed for estimating the expected Bayesian Q-values at all."
>
> We regret that our explanation was lacking in the first version. Our simulations and their discussion should now clearly reflect the following point: RL algorithms, like BEB and VBRB, output optimistic "estimates" or UCBs for the optimal Q-value. With the DMFP as reference, we can study just how loose this bonus is, or how biased.
>
> We have revised our numerical experiments to properly demonstrate this point.
>
> Also, we have run new experiments to first test the DMFP under different Dirichlet beliefs. In particular, that the beliefs reflect increasing sparsity of the MDP transition probabilities. This is done in two ways, as explained: either the prior becomes improper, or there is a true MDP which we can experience samples from, which then induces certain Dirichlet beliefs.
>
> ## Requested Changes: Major changes
>
> ### " It’s unclear why would we care about the expected Bayesian Q-values."
>
> As discussed above, we have put forth this argument more clearly now. Improved estimates of the moments of the optimal Q-values will assist in the design of better learning algorithms. In order to calculate the higher moments via the field theory, one first needs the first moment. Furthermore, the first moment already provides useful insights into existing RL algorithms.
>
> ### "The assumptions in this paper are not very clear. I would suggest the authors to focus on a single setting (e.g., when the mean reward is independent and normal) throughout the paper"
>
> We have clarified the assumptions now in section 2; both those of [Stamatescu2022] and the extensions we consider (mainly, non-identical beliefs). The independence assumption of rewards is maintained; indeed this is crucial for the mean field result.
>
> The simulations are now for Gaussian rewards with unknown mean, implying a conjugate prior/posterior being a Gaussian as well.  We thank the reviewer for the suggestion. Section 3 is still the general review of expected maxima, which in many cases apply to arbitrary distributions.
>
> ### "The comparison with BEB and VBRB in the numerical experiments is unfair."
>
> We regret our initial presentation did not explain this comparison. We describe much more clearly now that what we are identifying is how loose BEB and VBRB are. We can do this exactly in a certain cases, at the "start of learning", ie, under the prior beliefs. This is because if the mean field theory holds, the variance (and higher moments) of the Q-values is that of the mean-rewards.
>
> In the case of VBRB, this algorithm adds exploration bonuses to the certainty equivalent Bellman equation (which they refer to as the "mean MDP", but which is in fact a lower bound via Jensen's inequality). We see in Section 5 how loose the resulting optimistic Q-value is.

---

> > ### Author Response · Authors · 2023-09-22
> > **Response (continued)**
> >
> > ## Requested Changes: Minor changes
> >
> > All minor changes have been addressed in the revision, we thank the reviewer for their close scrutiny. We respond below for further clarification on certain points.
> >
> > ### I don’t see how “prior” and “posterior” make a difference in the DMFP updates, as throughout the paper the distributions of P and r serve as a background and the computation and updates of P and r are not discussed.
> >
> > We have removed this algorithm pseudocode as it was not particularly helpful.
> >
> > We hope our discussion and simulations are sufficiently clear now: the DMFP equations are exact under certain conditions, eg. under certain priors or posteriors. Outside of these conditions, such as 1) when we have an improper prior for the Dirichlet (all $\alpha_{s,a,s'}\to 0$) or, 2) when we have interacted with a sparsely connected MDP, the DMFP equations form an approximation (in the style of a message passing algorithm, but for RL).
> >
> > ### It would be better to clearly state the large state space limit assumption in a rigorous manner.
> >
> > We have more clearly stated the assumptions of the mean field theory. The large $N$ limit must be taken for the asymptotic independence of Q-values to hold, as we state. However, we clarify that a theorem has not been proven, the result is at the level of physics rigour. Work is underway on a rigorous proof.
> >
> > ### In fact, this seems to suggest that if we want to execute the policies from the posterior mean of the Bayesian Q-values, it only depends on the instantaneous/one-step reward means.
> >
> > Your reading is correct, but this only holds under the mean field conditions. Under the conditions, the Q-values become independent, so the agent beliefs they face a set of bandit problems (or a contextual bandit, with the state index as the context). However, outside of the mean field setting, the equations will most definitely not have a uniform constant across all state-actions.
> >
> > ### In Section 5, maybe I missed something but why Afonja, etc. are not compared (but only see “Ross’)?
> >
> > We have focused on the numerical integration a la Ross because it agrees with Afonja closely. In the case that we have 2 actions and these are Gaussian, it is of course easier and cheaper to use the exact expressions derived by Afonja, since these only require calling error functions.
> >
> >
> >
> > We thank the reviewer again for their time and effort.

---

### Decision · Action_Editor_DCNc · 2023-12-03

**Recommendation:** Reject

**Comment:**

Following the majority of reviewers, the paper cannot be accepted in its current form in TMLR. The main objections are related to the level of rigor of the writing, the lack of contextualization and clarity of the hypotheses and numerical experiments. I recommend the authors to take into full consideration the reviewers comments and consider a re-submission after improvement, especially given that it seems clear that the targeted audience is appropriate; the paper fall under the scope of TMLR.

**Audience:**

Concerning the audience, two reviewers over the three believe that the paper could potentially capture the attention of some readers of TMLR. I agree with this opinion.

**Claims And Evidence:**

Despite the revision, two among the three reviewers are not convinced by the supporting evidence. In particular, recurrent concerns relate to the lack of clarity of the hypotheses assumed in the work and for the analysis to hold (essentially an overall lack of rigour), as well as a lack of explanations of the actual meaningfulness of estimating the average only of the Q-values. Another issue relates to the lack of explanations for the numerical experiments and of fair comparisons with existing approaches. Related to the last point, the writing has not convinced some reviewers. It seems that the motivation for the work was not put enough in perspective and compared with exiting literature, a comment shared by the most positive reviewer. The revised version of the paper have partially adressed some of the concerns, in particular the last one. However, the authors should make a greater effort at motivating the work in a broader context and better explain the numerical parts as well as explain why such comparisons are meaningful.

**Resubmission Of Major Revision:**

The authors may consider submitting a major revision at a later time.